# Hsp40s play complementary roles in the prevention of tau amyloid formation

Rose Irwin[1†‡§], Ofrah Faust[1†], Ivana Petrovic[1#], Sharon Grayer Wolf[2], Hagen Hofmann[1], Rina Rosenzweig[1*]

[1]Department of Chemical and Structural Biology, Weizmann Institute of Science, Rehovot, Israel; [2]Department of Chemical Research Support, Weizmann Institute of Science, Rehovot, Israel

**Abstract** The microtubule-associated protein, tau, is the major subunit of neurofibrillary tangles associated with neurodegenerative conditions, such as Alzheimer's disease. In the cell, however, tau aggregation can be prevented by a class of proteins known as molecular chaperones. While numerous chaperones are known to interact with tau, though, little is known regarding the mechanisms by which these prevent tau aggregation. Here, we describe the effects of ATP-independent Hsp40 chaperones, DNAJA2 and DNAJB1, on tau amyloid-fiber formation and compare these to the small heat shock protein HSPB1. We find that the chaperones play complementary roles, with each preventing tau aggregation differently and interacting with distinct sets of tau species. Whereas HSPB1 only binds tau monomers, DNAJB1 and DNAJA2 recognize aggregation-prone conformers and even mature fibers. In addition, we find that both Hsp40s bind tau seeds and fibers via their C-terminal domain II (CTDII), with DNAJA2 being further capable of recognizing tau monomers by a second, distinct site in CTDI. These results lay out the mechanisms by which the diverse members of the Hsp40 family counteract the formation and propagation of toxic tau aggregates and highlight the fact that chaperones from different families/classes play distinct, yet complementary roles in preventing pathological protein aggregation.

**\*For correspondence:**
rina.rosenzweig@weizmann.ac.il

[†]These authors contributed equally to this work

**Present address:** [‡]Program in Molecular Medicine, Hospital for Sick Children, Toronto, Canada; [§]Department of Biochemistry, University of Toronto, Toronto, Canada; [#]Biozentrum, University of Basel, Basel, Switzerland

## Introduction

Tau is an intrinsically disordered protein (IDP) that is highly expressed in neurons and plays essential roles in microtubule self-assembly and stability (*Mandelkow and Mandelkow, 2012*), axonal transport (*Gustke et al., 1994*), and neurite outgrowth (*Biernat and Mandelkow, 1999*). Tau binds to microtubules via its central microtubule-binding repeat (MTBR) domain (*Figure 1a*), an interaction that is modulated by post-translational modifications (PTMs). Aberrant PTMs such as hyperphosphorylation and acetylation (*Hanger et al., 2009*; *Morris et al., 2015*; *Cook et al., 2014*) were suggested to decrease the affinity of tau to microtubules, thus subsequently reducing microtubule stability. In addition, when tau dissociates from microtubules, it can form oligomers with the potential to disrupt cellular membranes, thereby impairing synaptic and mitochondrial functions, before ultimately forming amyloid fibers (*Shafiei et al., 2017*).

These abnormal forms of tau are thought to play a key role in the pathogenesis of various human tauopathies, including Alzheimer's disease (AD), frontotemporal dementias, and progressive supranuclear palsy (*Ballatore et al., 2007*). In such cases, tau forms large intracellular aggregates, termed neurofibrillary tangles, whose abundance and localization in the brain correlates with cognitive decline (*Ballatore et al., 2007*; *Brunello et al., 2020*). It is still unclear, however, if the fibrils themselves are the neurotoxic species or whether prefibrillar soluble aggregates and oligomers of tau promote neuronal death by spreading tau pathogenicity from cell to cell in a prion-like manner.

Chaperone machineries, and in particular members of the Hsp70 and the ATP-independent, small heat shock protein (sHSP) families, engage with tau during these pathogenic events, counteracting

**eLife digest** Several neurological conditions, such as Alzheimer's and Parkinson's disease, are characterized by the build-up of protein clumps known as aggregates. In the case of Alzheimer's disease, a key protein, called tau, aggregates to form fibers that are harmful to neuronal cells in the brain. One of the ways our cells can prevent this from occurring is through the action of proteins known as molecular chaperones, which can bind to tau proteins and prevent them from sticking together.

Tau can take on many forms. For example, a single tau protein on its own, known as a monomer, is unstructured. In patients with Alzheimer's, these monomers join together into small clusters, known as seeds, that rapidly aggregate and accumulate into rigid, structured fibers. One chaperone, HSPB1, is known to bind to tau monomers and prevent them from being incorporated into fibers. Recently, another group of chaperones, called J-domain proteins, was also found to interact with tau. However, it was unclear how these chaperones prevent aggregation and whether they bind to tau in a similar manner as HSPB1.

To help answer this question, Irwin, Faust et al. studied the effect of two J-domain proteins, as well as the chaperone HSBP1, on tau aggregation. This revealed that, unlike HSBP1, the two J-domain proteins can bind to multiple forms of tau, including when it has already aggregated in to seeds and fibers. This suggests that these chaperones can stop the accumulation of fibers at several different stages of the aggregation process. Further experiments examining which sections of the J-domain proteins bind to tau, showed that both attach to fibers via the same region. However, the two J-domain proteins are not identical in their interaction with tau. While one of them uses a distinct region to bind to tau monomers, the other does not bind to single tau proteins at all.

These results demonstrate how different cellular chaperones can complement one another in order to inhibit harmful protein aggregation. Further studies will be needed to understand the full role of J-domain proteins in preventing tau from accumulating into fibers, as well as their potential as drug targets for developing new treatments.

its aggregation into amyloids and targeting the misfolded species for degradation (*Dou et al., 2003*; *Voss et al., 2012*; *Petrucelli et al., 2004*; *Mok et al., 2018*; *Caballero et al., 2021*). Moreover, in various cellular models, increased chaperone levels were shown to play an important role in tau cellular homeostasis by promoting tau solubility and microtubule binding while reducing the levels of pathological tau species (*Dou et al., 2003*; *Perez et al., 1991*; *Renkawek et al., 1994*; *Dabir et al., 2004*; *Sahara et al., 2007*; *Jinwal et al., 2013*; *Shimura et al., 2004*). The chaperones do so by specifically recognizing the six-residue aggregation-prone regions located at the start of the second and third repeats in the MTBR domain (*Mok et al., 2018*; *Freilich et al., 2018*; *Baughman et al., 2018*; *Weickert et al., 2020*; *Figure 1A*). These two motifs, $^{275}$VQIINK$^{280}$ and $^{306}$VQIVYK$^{311}$ (also called PHF6* and PHF6, respectively), are susceptible to the formation of β-sheet structures that have been found to be a prerequisite for tau aggregation (*von Bergen et al., 2000*; *Mukrasch et al., 2005*).

Despite both Hsp70 and HSPB1 chaperones recognizing the same regions, their tau aggregation-prevention mechanisms have been found to be markedly different. Hsp70 interaction with tau suppresses the formation of aggregation-prone tau nuclei and sequesters tau oligomers and fibrils (*Baughman et al., 2018*; *Patterson et al., 2011*), thereby neutralizing their ability to damage membranes and seed further tau aggregation (*Kundel et al., 2018a*). In contrast, HSPB1 was shown to delay tau fiber formation by weakly interacting with early species in the aggregation reaction (*Baughman et al., 2018*). Thus, it would appear that different cellular chaperones can bind to distinct tau species and affect tau homeostasis in different ways.

Excitingly, though, HSPB1 is not the only ATP-independent chaperone reported to interfere with the tau aggregation pathway – members of the Hsp40 family (also known as J-domain proteins [JDPs]) have also recently been reported to affect tau aggregation in the cell (*Mok et al., 2018*; *Brehme et al., 2014*; *Fontaine et al., 2015*; *Hou et al., 2020*).

JDPs are a diverse group of proteins that function as co-chaperones of the Hsp70 machinery, and are responsible both for selecting and delivering clients to the chaperones, and stimulating Hsp70

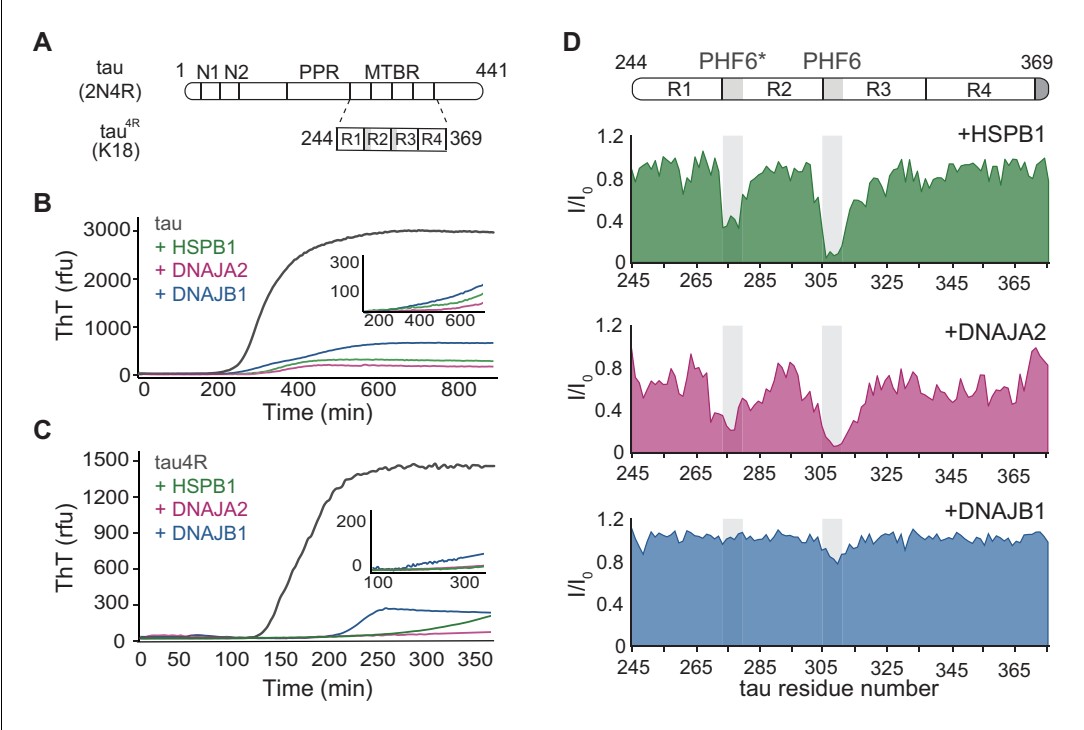

**Figure 1.** Interaction of chaperones with monomeric tau. (A) Domain organization of the longest splice isoform of tau protein (2N4R) and the short variant (4R/K18) containing only the microtubule binding repeats (MBTRs). The location of the two N-terminal inserts (N1 and N2) and the polyproline region (PPR) is indicated. The MTBR region of tau consists of four partially repeated sequences, R1 to R4, with the PHF6* and PHF6 aggregation-driving hexapeptides highlighted in gray. (B, C) ThT-based aggregation assay of 2N4R (B) and 4R (C) tau variants (10 µM) in the presence of 5 µM HSPB1 (green), DNAJA2 (purple), or DNAJB1 (blue) chaperones. The inset shows tau aggregation profiles in the presence of twofold excess (20 µM) of the same chaperones, showing complete inhibition over the course of the experiment. Representative data from three independent experiments is shown. (D) Tau$^{4R}$ binding profiles to HSPB1 (green), DNAJA2 (purple), and DNAJB1 (blue) chaperones, probed by NMR. Changes in NMR intensity ratios (I/I$_0$) upon addition of twofold excess of each chaperone are plotted as a function of tau$^{4R}$ residue number. The gray boxes represent the positions of the tau PHF6* and PHF6 aggregation-prone motifs. Values lower than 0.5 indicate intermolecular interactions.

The online version of this article includes the following source data and figure supplement(s) for figure 1:

**Source data 1.** Aggregation assay of 2N4R and 4R tau variants.

**Source data 2.** NMR binding experiments of tau$^{4R}$ to HSPB1, DNAJB1, and DNAJA2 chaperones.

**Figure supplement 1.** Interaction of chaperones with monomeric tau.

ATPase activity, thereby activating the chaperone cycle. These multidomain proteins are all structurally characterized by the conserved signature J-domain, essential for stimulation of Hsp70 ATPase activity (*Kityk et al., 2018*). In addition, canonical class A and B JDPs also comprise a regulatory glycine-rich (GF) region adjacent to the N-terminal J-domain (*Faust et al., 2020*; *Karamanos et al., 2019*), two structurally similar C-terminal β-barrel domains (CTDI and CTDII) containing the substrate binding region, and a dimerization domain (*Kampinga and Craig, 2010*; *Rosenzweig et al., 2019*). Class A JDPs further contain a zinc-finger-like region (ZFLR) protruding from CTDI.

Recently, several studies have indicated that JDPs can also function as bona fide chaperones, utilizing holdase activity to prevent the aggregation of their client proteins (*Ayala Mariscal and Kirstein, 2021*).

DNAJA2, a member of this JDP family, was recently identified as a potent suppressor of tau aggregation, capable of effectively preventing the seeding of tau and formation of amyloids in cells (*Mok et al., 2018*; *Abisambra et al., 2012*), with DNAJA2 levels being selectively increased in AD patient neuronal cells (*Mok et al., 2018*). Additionally, it was recently shown that, along with the Hsp70 system, DNAJB1, a class B JDP, can break apart tau amyloid fibers extracted from AD brain tissues (*Nachman et al., 2020*).

Little is known, however, regarding how these co-chaperones interact with tau or the mechanism by which they modify tau disease-related amyloid states.

We therefore used NMR spectroscopy, in combination with kinetic aggregation assays, to elucidate the effect of the DNAJA2 and DNAJB1 chaperones on tau aggregation. We found that the aggregation-prevention mechanisms of DNAJB1 and DNAJA2 are strikingly different from that of HSPB1 holdase chaperone. Moreover, we found that the two Hsp40 family members also diverge in their interactions with tau, whereas DNAJA2 interacts with all species along the tau aggregation pathway, including inert tau monomers, DNAJB1 only interacts with aggregation-prone tau conformers, such as seeding competent species or mature fibers.

## Results

### The chaperones slow down tau aggregation and bind tau

We first investigated the effect of DNAJA2 and DNAJB1 on the fibril formation of full-length tau (2N4R) and compared it to that of HSPB1, a well-characterized suppressor of tau aggregation. Tau aggregation was monitored using Thioflavin-T (ThT) fluorescence (*Biancalana and Koide, 2010*). The 3D GXG variant of HSPB1 (22) was used to mimic the fully activated dimeric form of the chaperone. As expected, the addition of HSPB1 significantly inhibited tau fiber formation, in agreement with previous reports (*Mok et al., 2018*; *Freilich et al., 2018*; *Baughman et al., 2018*). Interestingly, addition of DNAJA2 and DNAJB1 chaperones also completely inhibited the formation of tau fibrils for over 16 hr (*Figure 1B*), with no observable ThT signal being detected over this length of time. Similar results were obtained with the shorter tau construct tau$^{4R}$ (residues 244–372) (*Figure 1C*), which forms the core of tau filaments and nucleates tau aggregation (*Barghorn et al., 2004*). Despite the faster aggregation kinetics of tau$^{4R}$ relative to the full-length tau (t½ ≈ 110 min vs. t½ ≈ 340 min), the presence of even a sub-stoichiometric concentration of either of the three molecular chaperones (DNAJB1, HSPB1, or DNAJA2) fully suppressed aggregation for the duration of the experiment (*Figure 1C*).

As all three chaperones are able to suppress tau aggregation, we next aimed to understand the mechanisms by which they do so.

In order to unravel the mechanism of aggregation prevention, we first identified the binding sites for the three chaperones on tau. To this end, we recorded $^1$H-$^{15}$N HSQC spectra of either $^{15}$N-tau or $^{15}$N-tau$^{4R}$ in the absence and presence of each chaperone. Upon addition of HSPB1, we observed significant peak broadening of tau residues 275–280 and 306–311, which correspond to the PHF6* and PHF6 motifs (*Figure 1D* and *Figure 1—figure supplement 1*), in agreement with previous reports (*Mok et al., 2018*; *Freilich et al., 2018*; *Baughman et al., 2018*). Notably, the PHF6 motifs, which are the most hydrophobic regions within tau, are also the preferential binding sites for the major ATP-dependent chaperone families, such as Hsp70 and Hsp90 (*Mok et al., 2018*). A similar preference for the PHF6 and PHF6* motifs was also found for DNAJA2, with residues 275–284 and 306–320 showing significant peak broadening (*Figure 1D* and *Figure 1—figure supplement 1*). Surprisingly, however, DNAJB1 showed no significant binding to tau, despite efficiently suppressing amyloid formation (*Figure 1D* and *Figure 1—figure supplement 1*).

### Binding to tau fibrils

As DNAJB1 efficiently prevented tau aggregation in both full-length and tau$^{4R}$ experiments, yet showed no detectable binding to the monomers, we hypothesized that it functions, instead, through association with preformed tau fibers. In fact, a similar behavior was recently reported for DNAJB1 in the case of α-synuclein, where the chaperone displayed a remarkable preference (>300-fold) toward the amyloid state of α-synuclein over the monomer (*Wentink et al., 2020*).

We therefore checked whether DNAJB1 interacts with preformed tau$^{4R}$ fibrils using co-sedimentation experiments. Indeed, a large portion of DNAJB1 was detected in the insoluble fraction together with tau$^{4R}$ (*Figure 2A*), indicating a strong interaction between tau fibrils and the chaperone. Co-sedimentation experiments with DNAJA2 and HSPB1 showed that also DNAJA2 co-precipitated with tau fibers, whereas HSPB1 was mainly found in the soluble fraction and only marginally interacted with tau amyloids (*Figure 2A*). Similar results were also obtained with fluorescence anisotropy, resulting in an affinity (dissociation constant) for tau$^{4R}$ fibrils of 1.7 ± 0.2 μM (mean ± s.e.m.) for

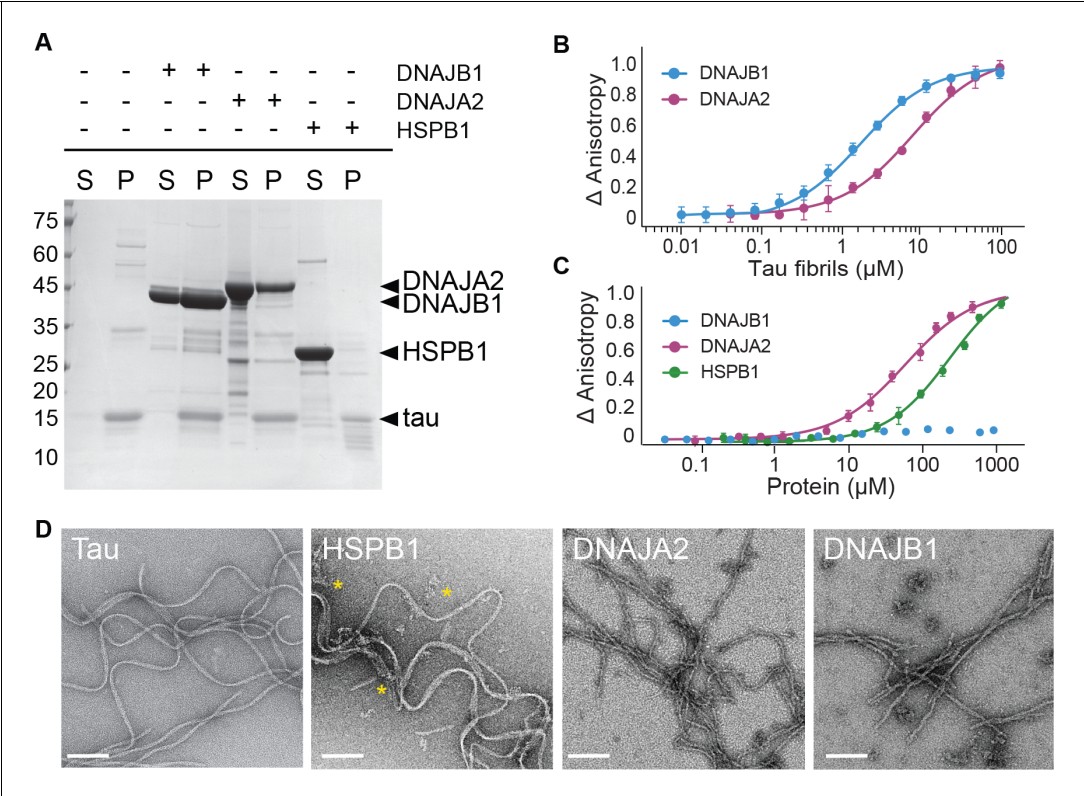

**Figure 2.** Interaction of chaperones with preformed tau fibrils. (**A**) Co-sedimentation of preformed tau amyloid fibrils (10 μM) in the presence of 10 μM DNAJB1, DNAJA2, or HSPB1 chaperones. SDS-PAGE of the supernatant (S) and pellet (P) fractions following ultracentrifugation is shown. (**B**) Fluorescence anisotropy assays of tau fibers binding to DNAJB1 (blue) or DNAJA2 (purple). Data points represent the means of three independent measurements (with standard deviations), with the $K_D$s for DNAJB1 and DNAJA2 being 1.7 ± 0.2 μM and 7.6 ± 0.6 μM, respectively. (**C**) Fluorescence anisotropy assays of monomeric tau binding to HSPB1 (green), DNAJB1 (blue), or DNAJA2 (purple). The $K_D$s were calculated as 250 ± 26 μM for HSPB1 and 43 ± 8 μM for DNAJA2. No binding was observed for DNAJB1. Data are mean ± s.e.m. (n = 3). (**D**) Representative negative stain electron microscopy micrographs of preformed tau fibers alone or upon incubation with 1:1 molar ratio of HSPB1, DNAJA2, or DNAJB1 chaperones. The asterisks show the position of HSPB1 chaperone clusters. White bar is 200 nm.

The online version of this article includes the following source data for figure 2:

**Source data 1.** Interaction of chaperones with preformed tau fibrils.

DNAJB1 and of 7.6 ± 0.6 μM for DNAJA2 (*Figure 2B*). In contrast, the affinity of HSPB1 for tau fibers was outside our experimental concentration range.

The selective interaction of DNAJB1 and DNAJA2 chaperones with tau fibers was also observed by negative stain electron microscopy (EM). Here, tau alone formed characteristic long twisted filaments with a periodicity of 50–100 nm (*Figure 2D*), consistent with previous observations (*Fitzpatrick et al., 2017*). Upon addition of HSPB1 chaperone to the fibers, no changes to fiber length or morphology were detected, in agreement with our co-sedimentation assays that showed no binding of HSPB1 chaperone to the preformed fibers. HSPB1 itself, however, generated large protein assemblies that can be seen next to the fibers in the EM images (marked by *).

Addition of DNAJB1 or DNAJA2, on the other hand, caused visible change in the appearance of the fibers, generating straighter, less twisted filaments decorated by periodically bound chaperones (*Figure 2D*). The overall length of the fibers, however, did not change substantially, and also smaller tau fragments were not observed in our EM images, indicating that DNAJB1 and DNAJA2 do not enhance fiber breakage or fragmentation upon binding. In summary, HSPB1 only binds tau[4R] monomers, DNAJB1 interacts with tau[4R] fibrils, whereas DNAJA2 binds both monomers and fibers (*Figure 2B, C*).

## Each chaperone suppresses a specific subset of microscopic processes in the tau aggregation reaction

We were next interested to see how the different tau binding modes of the three chaperones affect their respective aggregation-prevention mechanisms. We therefore performed a series of aggregation kinetics experiments, varying the concentrations of the chaperones while maintaining a constant concentration of tau$^{4R}$.

In the absence of chaperones, the aggregation of tau$^{4R}$ has previously been reported to occur through the following microscopic steps: primary nucleation, fibril growth through the addition and rearrangement of monomers (saturating elongation), and fiber fragmentation (*Kundel et al., 2018b*; *Yao et al., 2020*; *Figure 3—figure supplement 1*). To confirm that this is indeed the aggregation mechanism for tau$^{4R}$ under our experimental conditions, we recorded aggregation kinetics data at various concentrations of monomeric tau$^{4R}$ (2.5–40 µM; *Figure 3—figure supplement 1*). The half-saturation times (half-times) were then extracted and analyzed as a function of the concentration of tau$^{4R}$ monomer in accordance with a protocol by *Meisl et al., 2016*, allowing us to calculate the scaling exponent. This was found to be −0.34 ± 0.04, which was consistent with a dominant primary nucleation pathway and a contribution stemming from the presence of fibril fragmentation (*Kundel et al., 2018b*; *Yao et al., 2020*; *Shammas et al., 2015*; see Materials and methods for more detail). In addition, a positive curvature of these double-logarithmic plots indicated the presence of saturation effects in the dominant mechanism (*Meisl et al., 2016*; *Figure 3—figure supplement 1A*).

Tau$^{4R}$ aggregation kinetics data in the absence (*Figure 3—figure supplement 1B*) and presence of aggregation seeds (*Figure 3—figure supplement 1C*) were next globally fit, assuming a nucleus size of two tau$^{4R}$ monomers, to extract the kinetic rates for nucleation, elongation, and fragmentation, as well as the saturation constant (*Figure 3—figure supplement 1D*).

All chaperones caused a concentration-dependent retardation of tau aggregation at sub-stoichiometric concentrations (*Figure 3*). To understand the effect of the chaperones at the microscopic level, we fit the data to the kinetic model of tau aggregation using the kinetic parameters derived in the absence of chaperones. In order to determine which step in the aggregation mechanism is most likely affected by each of the chaperones, we only allowed an individual kinetic rate to vary in each of the analyses (see Materials and methods for detail). As the addition of any of the chaperones to preformed tau fibrils did not change the overall fibril length (*Figure 2D*), we could further assume that the chaperones have no effect on tau fragmentation rates ($k_m$).

Upon fitting HSPB1 aggregation-prevention data, only a poor agreement was achieved when allowing the perturbation of primary nucleation rates ($k_n$) (*Figure 3A*). Given the fact that HSPB1 binds tau monomers, the relatively moderate effect on tau primary nucleation was somewhat surprising as monomer binding should inhibit both nucleation and elongation. In contrast, fitting the fiber elongation rate ($k_p$) provided a significantly better description of the kinetic data (*Figure 3A*), indicating an order of magnitude reduction of this rate (*Figure 3D*). Moreover, even when allowing the variation of both the nucleation and elongation rates ($k_n$ and $k_p$) in the fit, only the elongation rates were reduced in the presence of HSPB1 chaperone, while the nucleation rates remained unchanged (*Figure 3—figure supplement 2A*, *Figure 3D*).

The effect of DNAJA2 on tau aggregation could neither be described by the reduction of nucleation rates nor elongation rates alone (*Figure 3B*), which is not entirely surprising, given that the chaperone can bind to both monomeric tau and fibrils (*Figure 1D*, *Figure 2B, C*). In general, DNAJA2 has at least two distinct pathways to impact aggregation, namely by (i) reducing the amount of monomeric tau accessible for nucleation and elongation and (ii) lowering the potency of fibrils to grow. In order to determine which rates are affected by the chaperone, we recorded an additional set of kinetic measurements in the presence of DNAJA2 chaperone, this time with the aggregation being initiated by tau fiber seeds. In the presence of seeds, tau primary nucleation events are negligible, thus allowing us to estimate the effect of the DNAJA2 chaperone only on rate constants of elongation and fragmentation (*Figure 3—figure supplement 2C*). Using a global fit of the unseeded and seeded aggregation kinetics, we were then able to determine the effect of DNAJA2 on the nucleation and elongation rates (*Figure 3—figure supplement 2C* and *Figure 3D*). Combined, these results show a substantial perturbation of both tau nucleation and elongation rates

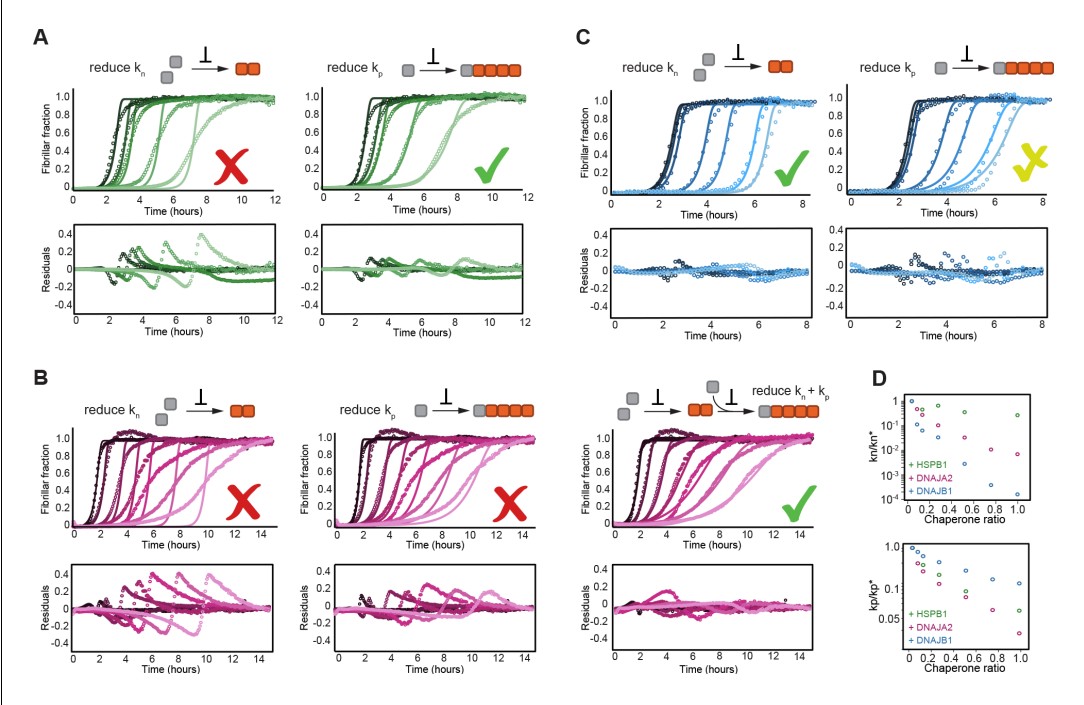

**Figure 3.** HSPB1, DNAJA2, and DNAJB1 chaperones affect different microscopic steps in the aggregation process. Kinetic profiles of the aggregation of 10 μM of tau in the absence (black) and presence of increasing concentrations of (A) HSPB1 chaperone (1, 2, 5, and 10 μM; dark to light green), (B) DNAJA2 (0.5, 1, 2.5, 5, 7.5, and 10 μM; dark to light purple), or (C) DNAJB1 (0.5, 1, 2.5, 5, 7.5, and 10 μM; dark to light blue). Open circles represent experimental data, and solid lines represent the fit of the kinetic profiles where only the primary nucleation ($k_n$, left) or elongation ($k_p$, right) pathways are inhibited. Residuals of the fits are shown under each panel. The changes in the aggregation kinetics caused by HSPB1 fit well with the elongation rate being primarily affected by the chaperone (A, right) whereas the changes in tau aggregation kinetics caused by DNAJB1 can be best described by the reduction of primary nucleation rates (C, left). In the case of DNAJA2, the changes in aggregation kinetics cannot be well described by the delay of only primary nucleation (B, left) or only elongation (B, middle) rates. There is good agreement, however, between the experimental data and fits to the integrated rate law of combined seeded and unseeded, in which primary nucleation, elongation, and fragmentation events have been considered simultaneously (B, right). Data points represent the means of 3–5 independent measurements. (D) The changes in microscopic nucleation (top) and elongation (bottom) rate constants as a function of the concentration of the molecular chaperones, relative to tau alone.

The online version of this article includes the following source data and figure supplement(s) for figure 3:

**Source data 1.** Kinetic profiles of tau aggregation in the presence of molecular chaperones.

**Figure supplement 1.** Global fitting of tau aggregation data.

**Figure supplement 2.** Effect of chaperones on tau aggregation kinetics.

by the DNAJA2 chaperone (*Figure 3D*), indicating that this chaperone affects the aggregation process in more intricate ways than HSPB1.

Surprisingly, the effect of DNAJB1 on tau aggregation was best described by its ability to reduce the rates of primary nucleation (*Figure 3C*, left) and, only to a lesser extent, fiber elongation (*Figure 3C*, right). This ability of the DNAJB1 chaperone to effectively inhibit the rate of tau[4R] primary nucleation was unexpected as primary nucleation involves interaction between tau monomers, yet no interaction between DNAJB1 and this species of tau was observed in our NMR experiments.

## Two conformations of monomeric tau

The effect of DNAJB1 on tau[4R] nucleation, despite its inability to interact with monomeric tau[4R], can, however, be reconciled by the recent finding that soluble monomeric tau[4R] exists in two conformational ensembles – an ensemble that does not spontaneously aggregate ('inert' tau monomer) and a seed-competent monomer that triggers the spontaneous aggregation of tau (*Mirbaha et al., 2018*; *Chen et al., 2019*). It is therefore possible that DNAJB1 only identifies and interacts with the 'aggregation-prone' tau species and not the inert monomers.

Such seed-competent species have been proposed to be readily populated in tauopathy-associated tau mutants (*Chen et al., 2019*), and can be generated in vitro by addition of polyanions such as heparin (*Eschmann et al., 2017*). Unfortunately, the addition of heparin causes the rapid formation of tau fibers, thus preventing a detailed structural investigation of the seed-competent ensemble using NMR spectroscopy.

Yet, whereas tau[4R] is inaccessible under conditions at which it rapidly aggregates, the rate of tau fibrillization can be tuned by altering the concentration of heparin present in the aggregation reaction. Low concentrations of heparin enhance the rate of fiber formation, whereas higher heparin concentrations potentially inhibit it (*Ramachandran and Udgaonkar, 2011*). We therefore performed aggregation kinetics at different heparin concentrations (0.1–40 µM) to identify the conditions at which tau aggregation is sufficiently slow to permit NMR experiments. We found that heparin increased the rate of fiber formation up to a sub-stoichiometric concentration of 1 µM (1:0.1 tau:heparin) (*Figure 4—figure supplement 1A*). Above this threshold, further addition of heparin in fact slowed tau aggregation in a dose-dependent manner. At a one- to fourfold excess of heparin (10–40 µM), we found that tau amyloid formation was arrested for over 2 hr (*Figure 4—figure supplement 1A*). An equimolar concentration of heparin to tau was therefore used to generate a soluble, aggregation-prone tau species (*Mirbaha et al., 2018*) that does not aggregate and is therefore amenable for NMR experiments.

We then monitored chaperone binding to this tau species by recording $^{1}$H-$^{15}$N HSQC spectra for $^{15}$N-tau-heparin complex alone, and upon addition of DNAJB1, DNAJA2, and HSPB1 chaperones to the mixture (*Figure 4*). To map the binding sites for the chaperones on this heparin-bound, aggregation-prone tau species, we first had to assign its spectrum. HNCA, CBCA(CO)NH, HN(CA)CO, and

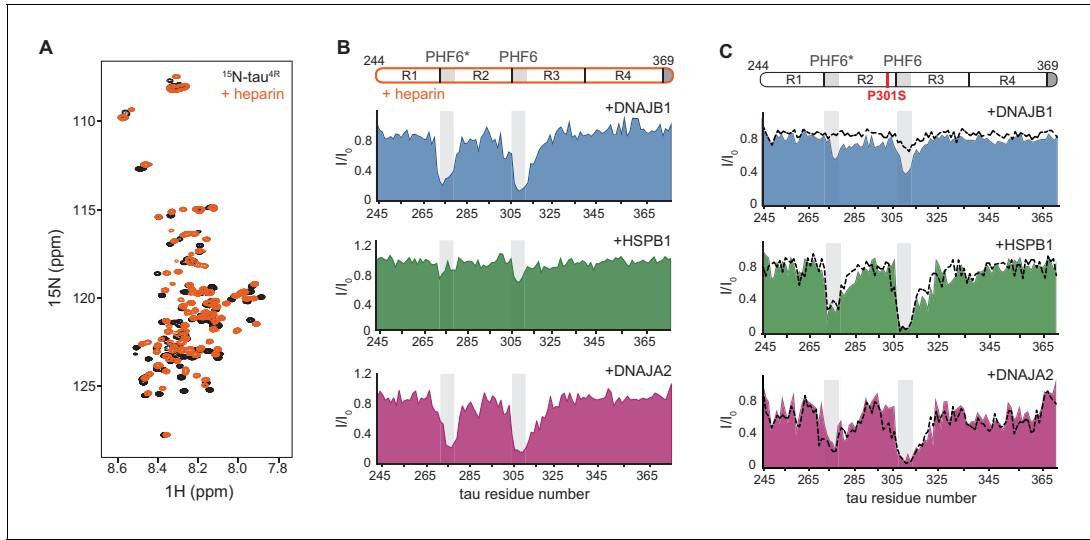

**Figure 4.** Interaction of chaperones with aggregation-prone tau species. (A) Overlay of two-dimensional $^{1}$H-$^{15}$N NMR spectra of tau in the absence (black) and presence of heparin (orange). Heparin binds mainly to the R1, R2, and PHF6* regions (see *Figure 4—figure supplement 1B*), with no changes observed to the overall dispersion of the spectra, indicating that heparin binding does not result in global folding of tau. (B) Residue-resolved NMR signal attenuation (I/I$_0$) of tau in complex with heparin upon addition of one molar equivalent of DNAJB1 (blue), DNAJA2 (purple), or two molar equivalents of HSPB1 (green). Gray boxes represent the positions of tau PHF6* and PHF6 aggregation-prone motifs in tau. Unlike in the case of tau monomer, DNAJB1 interacts strongly with this aggregation-prone species, while HSPB1 chaperone does not. (C) Residue-resolved backbone amide NMR signal attenuation (I/I$_0$) of tau P301S mutant upon addition of two molar equivalents of DNAJB1 (blue), HSPB1 (green), or DNAJA2 (purple) chaperones. Dashed lines represent the same intensity plots for wild-type tau (as in *Figure 1D*). Compared to the wild-type, P301S mutant shows increased interaction with DNAJB1.

The online version of this article includes the following source data and figure supplement(s) for figure 4:

**Source data 1.** NMR binding experiments of tau P301L mutant to HSPB1, DNAJB1, and DNAJA2 chaperones.

**Figure supplement 1.** Characterization of heparin-bound tau.

**Figure supplement 2.** Chaperone binding to tau tested at high ionic strength.

**Figure supplement 3.** Conformations of the various tau species measured through residual dipolar couplings (RDCs).

**Figure supplement 4.** Interaction of chaperones with tauopathy associated tau mutants.

HNCO 3D NMR experiments were recorded, and assignments were obtained for 88% of the non-proline residues. Heparin binding resulted in chemical shift perturbations (CSPs) to the NMR spectrum of tau$^{4R}$, mainly in the R1, R2, and PHF6* regions (*Figure 4A* and *Figure 4—figure supplement 1B*). No changes were observed to the overall dispersion of the spectrum, indicating that heparin binding does not induce global folding of tau, in agreement with previous findings (*Mukrasch et al., 2005*).

With these assignments in hand, we were able to identify that, despite having negligible affinity toward the tau monomer, the DNAJB1 chaperone indeed interacts strongly with the aggregation-prone tau-heparin mixture (*Figure 4B*), as we previously predicted. We further mapped this binding to tau residues 275–280 and 305–314 of the PHF6 and PHF6* repeats – the same regions to which both DNAJA2 and HSPB1 bind in the inert, monomeric tau.

We next tested whether DNAJA2 and HSPB1 interact with the heparin-bound tau. DNAJA2 showed strong binding to the R2 and R3 PHF6* repeats of this aggregation-prone form of tau$^{4R}$, similarly to its interaction with the free monomer (*Figure 4B*). In contrast, HSPB1 only bound this species weakly, despite previously displaying a strong interaction with free tau$^{4R}$. The lack of interaction between HSPB1 and aggregation-prone tau species explains our previous observation that HSPB1 does not affect the rate of fiber nucleation (*Figure 3A*), as such an inhibition would require a direct interaction with the aggregate nucleus or aggregation-prone tau.

Thus, while interacting with the same regions of tau, the three chaperones each display specific preferences for the different tau monomer conformers: HSPB1 interacts solely with tau monomers; DNAJB1 exclusively with the aggregation-prone species; and DNAJA2 with both.

It was unclear, however, how the chaperones discriminate between the inert form of tau and the aggregation-prone tau-heparin complex.

## Aggregation-prone tau species

Secondary-structure propensity analysis (*Marsh et al., 2006*) of free and heparin-bound tau demonstrated that the different chaperone binding profiles cannot be explained by heparin-induced formation of extended β-strand structures in the PHF6 repeats (*Eschmann et al., 2017*) as these regions displayed no increase in secondary-structure propensity upon addition of heparin (*Figure 4—figure supplement 1C*). Likewise, no notable changes were observed between the Cα and Cβ secondary chemical shifts patterns of heparin-bound and free tau samples (*Figure 4—figure supplement 1D*).

We then set out to determine whether the differences between monomeric tau$^{4R}$ and the heparin-tau complex are caused by electrostatic interactions. Heparin is a polyanion with a net charge of −3 per disaccharide at pH 7.0 (or estimated charge density of −6.7 e-/kD; *Lin et al., 2020*). In contrast, tau$^{4R}$ has a positive net charge of +9.5, suggesting that the heparin-tau$^{4R}$ complex is significantly stabilized by electrostatic attractions. As such, the ability of the chaperones to distinguish between the monomeric and aggregation-prone tau species may be related to simple differences in the net charge of the complex, with heparin binding, for example, reducing electrostatic repulsions that may exist between DNAJB1 and monomeric tau$^{4R}$. Screening any such electrostatic interactions with higher ionic strengths (0.3 M), however, neither facilitated binding between DNAJB1 and tau$^{4R}$ (*Figure 4—figure supplement 2*) nor altered tau$^{4R}$ interaction with HSPB1 or DNAJA2 chaperones (compare *Figure 1D* and *Figure 4—figure supplement 2*). Similarly, the interaction cannot be explained by attractive electrostatic interactions caused by the negative net-charge of the complex as we did not detect specific binding between heparin and any of the chaperones. Hence, electrostatic interactions cannot explain the different binding profiles of the chaperones to the two tau species.

A clue to how heparin alters the conformational ensemble of tau$^{4R}$ came from a recent structural characterization of patient-derived, seeding-competent tau monomer. This species of tau was reported to have an expanded ensemble with a more exposed PHF6 motif compared to the inert monomeric protein (*Mirbaha et al., 2018*; *Chen et al., 2019*). This increased expansion of the PHF6 aggregation motifs was, in turn, suggested to drive the self-assembly and subsequent aggregation of tau (*Kaufman et al., 2017*). Interestingly, binding of heparin to tau was also shown to expand the local conformation of the repeat regions (R2 and R3), thereby making the amyloidogenic PHF6 sequences more accessible (*Eschmann et al., 2017*).

Indeed, in agreement with previous reports (*Mukrasch et al., 2007*; *Mukrasch et al., 2009*), measurements of one-bond N-H RDCs in tau, which had been partially oriented in either a Pf1

bacteriophage or polyethylene glycol/hexanol alignment medium, showed relatively large H-N RDC values (10–20 Hz) in the PHF6 repeat regions, which can arise from a locally compacted conformation (*Figure 4—figure supplement 3A, B*). Similar analysis of the tau-heparin complex showed only very small RDCs in the PHF6 regions, despite the larger degree of alignment of this complex, indicating a potential expansion of these regions (*Figure 4—figure supplement 3C*). This expansion, and thus the increased accessibility of the PHF6 motif in the heparin-bound state, may therefore be what enables the interaction of DNAJB1 with tau.

To further test whether the expansion of the PHF6 repeats in the seeding-competent tau species is indeed the discriminating factor for chaperone binding, we monitored the interaction of the three chaperones with P301L/S missense mutations, which are known to cause dominantly inherited tauopathy (*Rizzu et al., 1999*). These tau variants had been shown to contain a higher population of the expanded PHF6 conformation (*Chen et al., 2019*), as also seen in our RDC measurements (*Figure 4—figure supplement 3D, E*), and could therefore potentially mimic the aggregation-prone tau without requiring the addition of heparin.

Similarly to what we observed for wild-type tau[4R], we found that indeed all three chaperones efficiently suppressed the aggregation of these tau mutants (*Figure 4—figure supplement 4B*), although with reduced efficacy in the case of HSPB1 and higher activity of DNAJB1 (*Figure 4—figure supplement 4C*).

The interaction of the P301L and P301S familial mutations of tau[4R] with HSPB1, DNAJB1, or DNAJA2 chaperones was then assayed using NMR spectroscopy. In the presence of HSPB1 and DNAJA2, the resulting binding profiles of the two tau variants were very similar to those of wild-type tau, with residues 274–280 and 305–318, corresponding to PHF6* and PHF6 repeats, displaying severe peak broadening (*Figure 4C* and *Figure 4—figure supplement 4A*). This similarity to wild-type tau was not unexpected, though, as only a small portion of the P301L/S tau conformational ensemble adopts the extended conformation at any given moment (*Chen et al., 2019*; *Kawasaki and Tate, 2020*). In contrast, DNAJB1, while showing no binding to wild-type tau, did cause noticeable reductions in peak intensities in both PHF6 motifs of the P301S and P301L tauopathy mutants (*Figure 4C* and *Figure 4—figure supplement 4A*). Moreover, the degree of observed intensity reduction (31 and 26%) was found to be consistent with the relatively low population of the expanded conformation in tau P301L/S mutants (*Kawasaki and Tate, 2020*; *Figure 4—figure supplement 3D and E*).

Thus, DNAJB1 binding to monomeric tau indeed appears to depend on the exposure of the PHF6 region, thereby allowing it to distinguish between inert tau and the aggregation-prone species that eventually leads to amyloid formation.

## Changes to tau fibers caused by the chaperones

Our results show that all three chaperones efficiently inhibit tau aggregation via interactions with the hydrophobic aggregation-prone PHF6 repeats. Each of the chaperones interacts with a specific set of tau species, thus slowing down different microscopic processes in the aggregation reaction (*Figure 5D*). However, it remained unclear whether the chaperones also affect the size and/or morphology of tau fibers.

Since ThT fluorescence only reports on total fibril mass, with no differentiation to fibril length or number, we turned to EM in order to image the fibers. When adding DNAJA2 at the start of the aggregation reaction, a clear dose-dependent decrease in the density and length of the tau fibrils was observed (*Figure 5A*). Similarly to DNAJA2, DNAJB1-containing reactions yielded shorter fibrils that further shortened when repeating the experiment with increasing chaperone concentrations (*Figure 5B*). This result is not unexpected given that these two chaperones also bind to tau[4R] fibers (*Figure 2*), which may contribute to the arrest of tau amyloid elongation.

Interestingly, DNAJA2 was very efficient in reducing both the size and amount of formed tau fibers, and even at the sub-stoichiometric concentration of 0.1:1 DNAJA2:tau, a significant reduction in fiber length was observed. At a ratio of 0.25:1 DNAJA2:tau, the majority of fibers were between 0.5 and 2.0 μm long, and at 0.5:1 DNAJA2:tau the fibers were too short to be detected (*Figure 5A*). These results were also in agreement with our kinetic measurements at 0.5:1 DNAJA2:tau ratio, which showed an 88% reduction in fibril mass (*Figure 5—figure supplement 1A*). Thus, the ability of DNAJA2 chaperone to interact with all tau species and to prevent both elongation and nucleation rates makes it a potent suppressor of tau aggregation. DNAJB1, which does not interact with tau

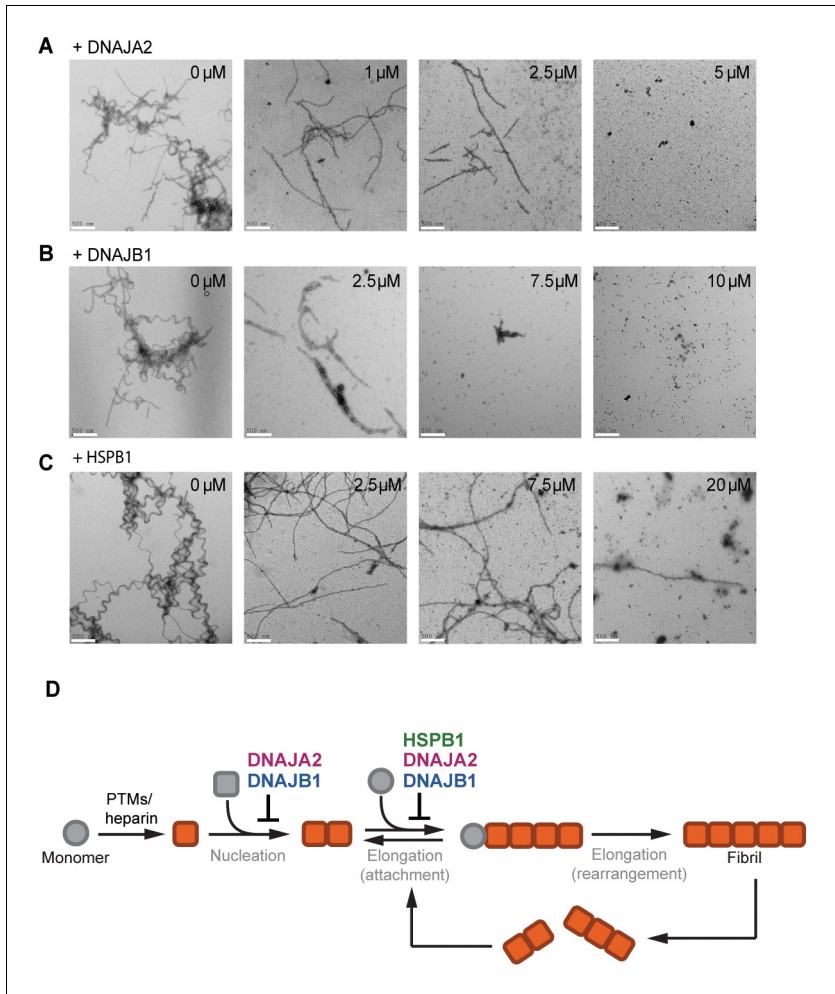

**Figure 5.** The chaperones change the morphology of tau amyloids. (A–C) Representative negative stain electron micrographs of end products of the tau aggregation-prevention assays preformed in the presence of indicated concentrations of DNAJA2 (A), DNAJB1 (B), and HSPB1 (C). White bar is 500 nm. Tau$^{4R}$ forms long fibrils in the absence of chaperones and is still able to form long, fully developed fibrils in the presence of HSPB1 dimer. It forms smaller oligomeric species in the presence of both DNAJA2 and DNAJB1, indicating that these chaperones maintain much of the protein in a non-fibrillary state. (D) Overview of the variety of diverse microscopic mechanisms through which HSPB1, DNAJB1, and DNAJA2 molecular chaperones can suppress tau amyloid formation, as revealed by the binding and kinetic assays in this paper. These results demonstrate that the chaperones have evolved to exploit the different opportunities to modulate tau aggregation by binding to different tau species along the aggregation pathway.

The online version of this article includes the following figure supplement(s) for figure 5:

**Figure supplement 1.** Effect of chaperones on tau fibril mass.

monomers, was less effective than DNAJA2 in reducing fiber size and mass (67% reduction; *Figure 5—figure supplement 1B*), and at 0.5:1 DNAJB1:tau the fibers were still visible. These were, however, significantly shortened, with sizes ranging from 0.2 to 1.0 µm, and at a stoichiometric ratio of 1:1 no fibers were observed (*Figure 5B*).

HSPB1 chaperone, on the other hand, affected fiber formation differently when added at the start of the tau aggregation reaction. While a clear dose-dependent decrease in the amount of the tau fibrils was observed (*Figure 5C* and *Figure 5—figure supplement 1C*), unlike DNAJB1 and DNAJA2, HSPB1 was unable to completely suppress the formation of fibers at sub-stoichiometric concentrations. Even in the presence of twofold excess of HSPB1 chaperone, a few fibers per micrograph were still observed, which, interestingly, were somewhat shorter and straighter in appearance

compared to the long twisted fibrils of tau alone (*Figure 5C*). Although the reduced elongation rate in the presence of HSPB1 decelerates the aggregation process, HSPB1 only has a limited effect on the final fiber mass, as also evident in our kinetic measurements, showing only 45% reduction in fibril mass even at equimolar HSPB1:tau concentrations. Thus, the weak interaction of HSPB1 with tau monomers along with its limited ability to only slow down the fibril elongation rate is not sufficient to efficiently prevent tau incorporation into the amyloid fibers.

Thus, DNAJB1 and DNAJA2 chaperones, which bind to both aggregation-prone tau species and fibers, are significantly more efficient in preventing the formation of mature fibers than HSPB1, which can only bind to tau monomers.

## Chaperone interactions with tau

Overall it appears that the DNAJB1 and DNAJA2 chaperones, despite being very similar in structure, display distinct differences in their interaction with tau. While DNAJA2 binds to all tau species – monomers, aggregation-prone tau, and fibers – DNAJB1 does not bind at all to inert tau monomers, but interacts strongly with the aggregation-prone species as well as mature fibers.

This disparity between the chaperones could be explained by utilization of different structural domains to recognize the various tau species; however, no structural information is currently available for either DNAJA2 or DNAJB1 in complex with tau[4R]. We therefore utilized NMR to map the binding sites on the chaperones for the various tau species.

Both DNAJA2 and DNAJB1 are homodimeric proteins comprising an eponymous N-terminal JD that is essential for Hsp70 activation, two putative substrate binding domains, CTDI and CTDII, and a C-terminal dimerization domain. In addition, DNAJA2, as all class A JDPs, has a ZFLR insertion in CTDI (*Kampinga and Craig, 2010*; *Ayala Mariscal and Kirstein, 2021*; *Jiang et al., 2019*), while DNAJB1 has an autoinhibitory GF region connecting the JD to CTDI and blocking premature interaction of Hsp70 with the JD (*Faust et al., 2020*).

Due to the large size of the DNAJA2 dimer (90 kDa), which hampers NMR experiments, we used a monomeric version of Ydj1, a DNAJA2 homologue from *Saccharomyces cerevisiae*, that contains only the substrate binding and ZFLR domains (*Li et al., 2003*). This construct (27 kDa) of $^2$H, $^{15}$N-labeled DNAJA2(Ydj1)$^{111-351}$ was far more amenable to NMR and, as previously reported, gave a high-quality $^1$H-$^{15}$N HSQC-TROSY spectrum (*Jiang et al., 2019*; *Figure 6—figure supplement 1A*). Addition of twofold excess of tau[4R] caused CSPs in the first DNAJA2 substrate-binding domain (*Li et al., 2003*), located in CTDI (*Figure 6A, C* and *Figure 6—figure supplement 1A*). Specifically, tau[4R] (most likely via the hydrophobic PHF6 and PHF6* motifs) binds to a hydrophobic pocket located between β-strands 1 and 2 (*Figure 6C*, colored purple). The binding was in fast exchange, and no reduction in peak intensities to DNAJA2(Ydj1)$^{111-351}$ residues was observed (*Figure 6—figure supplement 1A*), in agreement with the relatively low affinity of DNAJA2 for monomeric tau (43 µM, *Figure 2C*). DNAJB1, despite its CTDI having 56% identity to the CTDI tau-binding region of DNAJA2, showed no interaction with the monomeric tau (*Figure 6E* and *Figure 6—figure supplement 1E, F*), as previously seen from the tau side (*Figure 1C*).

We then repeated the binding experiment using aggregation-prone tau species, generated by the addition of heparin (*Eschmann et al., 2017*). Interestingly, this tau species, unlike the inert tau monomer, caused significant peak broadening to residues in the second substrate binding region of DNAJA2, located in CTDII (*Figure 6B* and *Figure 6—figure supplement 1B*), whereas only small changes were detected in the CTDI region (*Figure 6—figure supplement 1B*). Hence, the DNAJA2 substrate-binding groove in CTDI interacts predominantly with the inert, monomeric tau, while the aggregation-prone tau species preferentially binds to CTDII (*Figure 6C, D*). In order to verify that our binding results were not affected by the deletion of the dimerization domain in the DNAJA2(Ydj1)$^{111-351}$ construct, we then repeated the binding experiments using a construct comprising only CTDII and the Ydj1 dimerization domains, termed DNAJA2(Ydj1)$^{256-409}$. This DNAJA2 variant, as expected, showed no binding to monomeric tau in our NMR experiments, confirming that tau indeed binds to the CTDI region, which was lacking in this construct (*Figure 6—figure supplement 1C*). Addition of heparin-bound tau, however, caused significant CSPs to the spectrum of DNAJA2(Ydj1)$^{256-409}$ (*Figure 6—figure supplement 1D*), again confirming the binding of this aggregation-prone tau species to CTDII.

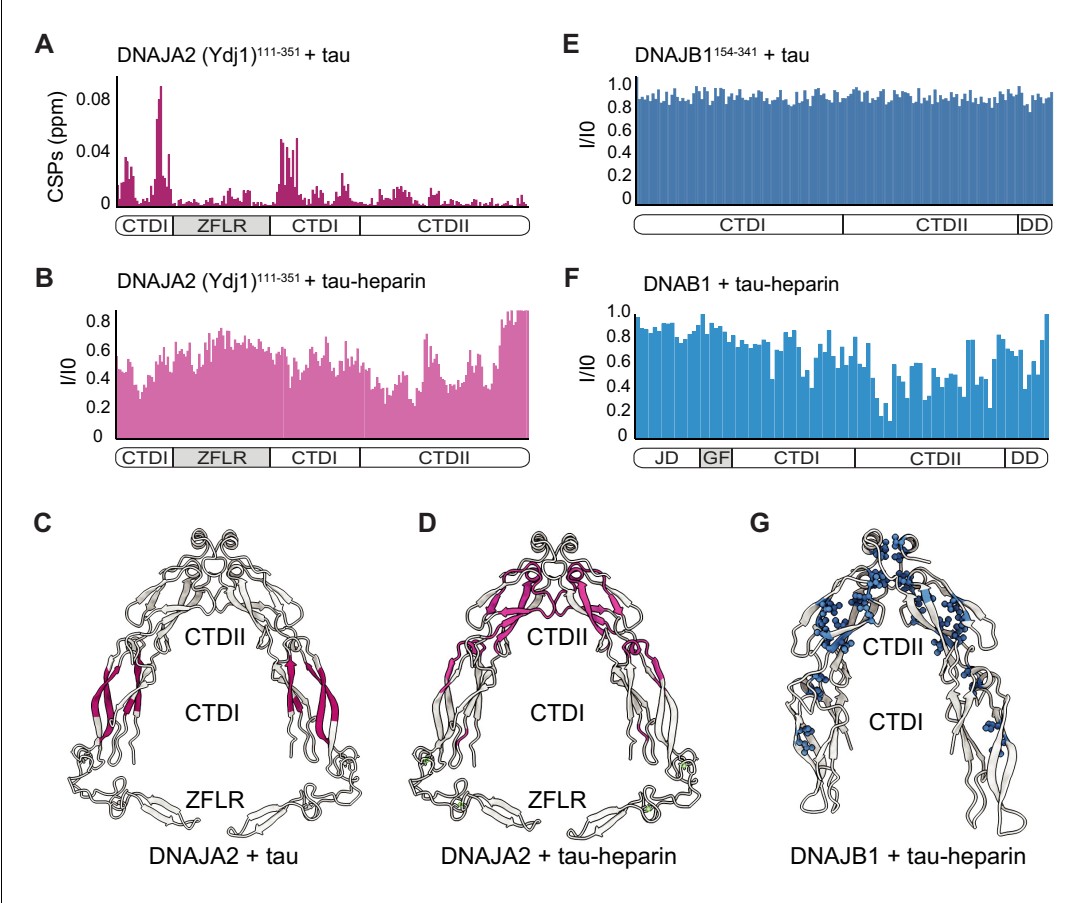

**Figure 6.** Mapping DNAJA2 and DNAJB1 chaperone binding to tau species. (**A**) Combined amide chemical shift perturbations between free DNAJA2 (Ydj1)$^{111-351}$ and DNAJA2(Ydj1)$^{111-351}$ bound to unlabeled tau. The domain organization of DNAJA2 is indicated at the bottom. Monomeric tau binds to the C-terminal domain I (CTDI) of DNAJA2, but not to the zinc-finger-like region (ZFLR) insertion. (**B**) Residue-resolved backbone amide NMR signal-attenuation (I/I$_0$) plot for DNAJA2(Ydj1)$^{111-351}$ in complex with aggregation-prone tau species, generated by addition of twofold excess heparin. The binding is primarily to the C-terminal domain II (CTDII) of DNAJA2. (**C, D**) Structural representation of DNAJA2 chaperone, with residues showing significant chemical shift perturbations (CSPs) upon binding to monomeric tau (from panel **A**) highlighted in purple (**C**), and residues displaying significant decreases in peak intensities upon binding to aggregation-prone tau species colored pink (**D**). (**E**) No changes in signal intensity are detected in the CTDs of DNAJB1 (residues 154–341) upon addition of twofold excess of monomeric tau protein, indicating a lack of interaction. (**F**) Combined methyl group chemical shift perturbations between free full-length DNAJB1 and DNAJB1 bound to aggregation-prone tau species generated by addition of twofold excess heparin. Binding is observed in the CTDII of DNAJB1. (**G**) Structural representation of DNAJB1 chaperone, with methyl residues showing significant changes upon binding to aggregation-prone tau species highlighted in blue.

The online version of this article includes the following figure supplement(s) for figure 6:

**Figure supplement 1.** Mapping DNAJA2 and DNAJB1 chaperone binding to tau species.

Having this second, distinct tau-binding domain thus explains both the high affinity of DNAJA2 toward the aggregation-prone tau and the ability of the chaperone to interact with the two tau species.

We then tested the binding of DNAJB1$^{154-341}$ (a dimeric protein lacking the N-terminal JD) to this aggregation-prone tau species. The interaction with tau-heparin, however, caused severe peak broadening, preventing us from obtaining site-specific information. The high molecular mass of the complex formed between multiple aggregation-prone tau monomers and the DNAJB1-dimer is presumably responsible for this effect. To overcome this problem, we utilized a $^2$H $^{13}$CH$_3$-ILVM sample of full-length DNAJB1 that gave a high-quality $^1$H-$^{13}$C HMQC spectrum even upon complex formation with the aggregation-prone tau (***Figure 6—figure supplement 1G***). Selective peak broadening was detected in methyl residues located in CTDII (***Figure 6F, G*** and ***Figure 6—figure supplement 1G***), indicating that, similarly to DNAJA2 chaperone, the aggregation-prone tau binds to CTDII in

DNAJB1. This CTDII site in DNAJB1 was also recently identified as a binding site for another amyloid-forming protein, α-synuclein (*Faust et al., 2020*).

Hence, DNAJB1 and DNAJA2 recognize seed-competent tau[4R] species predominantly via CTDII, yet the structural differences between the CTDI domains of the chaperones likely cause the diverging specificities for the various tau[4R] species.

## Discussion

Hsp70 chaperones are known to be key factors in tau quality control and turnover (*Miyata et al., 2011*); however, the contributions of their Hsp40 co-chaperones remain poorly understood. In this study, we describe the effects of Hsp40 chaperone family members, DNAJA2 and DNAJB1, on tau amyloid fiber formation in comparison to the well-characterized tau aggregation suppressor HSPB1.

DNAJA2 chaperone was previously identified as a potent suppressor of tau aggregation (*Mok et al., 2018*). Our results demonstrate that DNAJA2 can interact simultaneously with multiple tau species, which can explain its high effectiveness in aggregation prevention. DNAJA2 does not only bind tau monomers via its CTDI, thus preventing fiber growth by monomer addition, but it also binds aggregation-prone tau species via its CTDII, which effectively reduces the speed of nucleation.

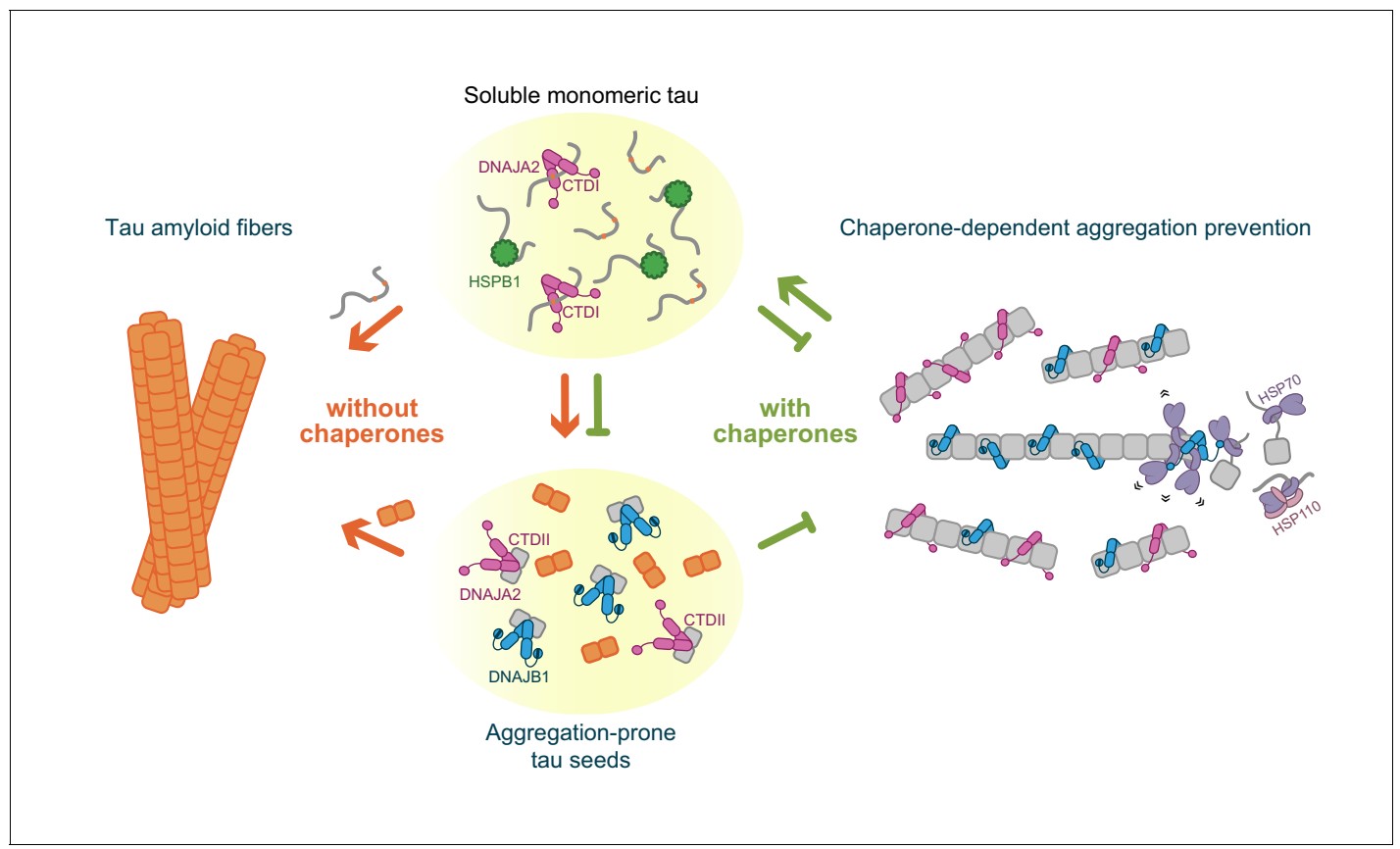

**Figure 7.** Schematic representation of chaperone-dependent aggregation prevention of tau. HSPB1, DNAJA2, and DNAJB1 chaperones play complementary roles in tau aggregation prevention, each affecting different microscopic steps along the tau aggregation pathway and interacting with distinct tau species. HSPB1 chaperone (green) interacts solely with monomeric tau (gray squiggles), preventing its incorporation into the growing fibers. DNAJB1 chaperone (blue) shows a completely distinct tau binding profile – binding only to aggregation-prone tau and to the mature tau fibers, thus inhibiting the formation of nuclei (orange squares), as well as the elongation of existing fibers. Furthermore, DNAJB1, together with the Hsp70 chaperone system, can disaggregate tau fibers, returning the proteins to their soluble monomeric state (*Nachman et al., 2020*; *Ferrari et al., 2018*). DNAJA2 (purple) binds to all tau species (monomer, nuclei, and fibers), thus serving as an extremely potent suppressor of tau aggregation. While nuclei and mature fibers selectively interact with a site in the C-terminal domain II (CTDII) of both Hsp40 chaperones, DNAJA2 possesses a second, distinct site in CTDI through which it recognizes the monomeric tau species.

Moreover, DNAJA2 even associates with mature tau fibers, which provides an additional pathway for inhibiting the incorporation of new monomers into the growing fibers (*Figure 7*).

Given this multiplicity of interfering pathways, DNAJA2 has the potential to serve as an early protective cellular factor that limits tau aggregation, which explains the correlation between distorted cellular DNAJA2 levels and pathology-linked nucleation sites (*Mok et al., 2018*).

Remarkably, in comparison to DNAJA2, DNAJB1 displayed significantly different interactions with tau despite the high structural similarity between both chaperones. In fact, unlike DNAJA2, DNAJB1 does not interact with inert tau monomers, although its CTDI is highly homologous to that of the class A chaperone. DNAJB1 affinity, however, is increased by orders of magnitude for the aggregation-prone tau species. There, DNAJB1 also functions as a bona fide chaperone, effectively hindering tau aggregation by preventing the formation of the seeding nuclei, as well as by stably binding to the amyloid fibers themselves, slowing down their growth (*Figure 7*). Interestingly, both interactions are mediated by CTDII, thus leaving the CTDI site free to potentially recruit Hsp70 chaperones and initiate the disaggregation of fibers when these are formed.

This mode of action of DNAJB1 differs significantly from that of the well-characterized tau aggregation-suppressor HSPB1, as well as from the recently described activity of the DNAJC7 chaperone (*Hou et al., 2020*). These chaperones preferentially bind to the PHF6 repeats in inert tau monomers, thus protecting these aggregation-prone regions and preventing them from being incorporated into the growing fibers. By solely interacting with inert tau monomers, though, HSPB1 can only effectively inhibit the rate of fibril elongation, and thus has a limited effect of preventing tau amyloid formation (*Figure 7*).

In contrast, the actions of DNAJA2 and DNAJB1 result in a significant decrease not only in fibril mass, but also length. This function of the Hsp40 chaperones, compared to HSPB1, could be attributed to their ability to interact with the forming tau fibers, which efficiently blocks the incorporation of additional tau monomers. In addition, the Hsp40 chaperones bind aggregation-prone tau species, thereby blocking their elongation into the mature tau fibers.

The relatively minor differences in the aggregation-prevention abilities of DNAJB1 and DNAJA2, along with the relatively poor performance of HSPB1 in preventing fibril growth, suggest that, while interaction with soluble tau monomers helps slow aggregation, this does not significantly contribute to the ability of chaperones to prevent amyloid growth once it has begun. It is therefore possible to envision a scenario in which HSPB1 delays amyloid formation during the early stages of tau aggregation via interaction with the monomers, and Hsp40 chaperones later interact with seeds and more mature species to further hinder the fibril formation process.

One open question is how these chaperones affect the aggregation of tau mutants linked to tauopathies. It has been hypothesized that some variants, such as P301L, may be capable of 'avoiding' the chaperone system, thus possibly contributing to the disease pathology. Such behavior was indeed recently observed with the DNAJC7 chaperone, which was found to have a significantly reduced affinity to the tau P301L mutation (*Hou et al., 2020*). Furthermore, in our aggregation-prevention assays, a reduction was observed in the activity of HSPB1 when incubated with the P301L variant compared to wild-type tau (*Mok et al., 2018*; *Figure 4—figure supplement 4B*). For these two chaperones, their reduced efficacy is likely due to the equilibrium of the P301L mutation shifting toward an aggregation-prone seeding conformation of tau (*Chen et al., 2019*), for which both DNAJC7 (30) and HSPB1 (*Figure 4B*) display reduced affinities. In contrast, DNAJA2 and DNAJB1 remained effective in suppressing the aggregation of the P301L variant of tau$^{4R}$ (with the anti-aggregation activity of DNAJB1 being even higher for this tauopathy mutant, *Figure 4—figure supplement 4C*), indicating that these Hsp40 chaperones could be effective in suppressing a wide range of tauopathies.

A second, crucial question that has yet to be answered is whether the reduction of fibril growth by the Hsp40 chaperones, which results in generation of smaller tau fibrils, is indeed beneficial for cellular homeostasis. A recent study showed that the disaggregation of tau by the DNAJB1/Hsp70/Hsp110 chaperones generates low-molecular-weight tau species, which were seeding-competent in cell culture models (*Nachman et al., 2020*). Hence, chaperone-mediated tau disaggregation may not be beneficial per se, but may instead be involved in the prion-like propagation of tau pathology. The smaller tau species generated during the DNAJA2 and DNAJB1 aggregation-prevention processes could then have similar prion-like propagation properties, acting as seeds that can sequester more tau protein into amyloid aggregates. In such a case, chaperone-mediated

aggregation prevention would, in fact, accelerate the progression of disease, ultimately proving detrimental to cell health. HSPB1 chaperones, on the other hand, do not generate smaller tau species during their aggregation prevention and could therefore be more beneficial in slowing the progression of disease, despite their lower chaperoning activity.

However, another important aspect to consider is that aggregation prevention in the cell does not occur in isolation and can also be coupled to protein degradation via the proteasome or autophagy. These pathways could potentially be more potent in degrading smaller tau fibrils and aggregation-prone monomers than the mature fibrils, thereby rendering the activity of DNAJA2 and DNAJB1 beneficial overall.

Hence, further studies will be required to understand the full role of Hsp40-mediated tau aggregation prevention in the cell, as well as to evaluate the therapeutic potential of the Hsp40 chaperone machineries in combating tauopathies.

## Materials and methods

### Construct preparation

Tau$^{4R}$ (residues 244–372, C322A) wt and mutants, HSPB1 (S15D, S78D, S82D, I181G, V183G), DNAJA2, and DNAJB1 (residues 154–341) were expressed in *Escherichia coli* BL21 (DE3) cells from pET-29b(+) vector with a N-terminal His$_6$ tag followed by a tobacco etch virus (TEV) protease cleavage site. Tau (C322S), DNAJB1, and Ydj1 (yeast orthologue of DNAJA) constructs were expressed from the pET-SUMO vector with an N-terminal His$_6$ purification tag and a Ulp1 cleavage site (DNAJB1 plasmid was a gift from B. Bukau, University of Heidelberg).

### Protein expression

Cells were grown in Luria Bertani broth (LB) to OD$_{600}$ ≈ 0.8 at 37°C, and expression was induced by addition of 1 mM isopropyl-β-D-thiogalactoside (IPTG). Cells expressing HSPB1, DNAJA2, and DNAJB1 chaperone variants were allowed to proceed overnight at 25°C and cells expressing tau constructs at 18°C.

Isotopically labeled tau and tau$^{4R}$ proteins for NMR were grown in M9 H$_2$O media supplemented with $^{15}$NH$_4$Cl (and $^{13}$C-glucose) as the sole nitrogen (and carbon) source. Protein expression was induced with 1 mM IPTG at 18°C overnight.

Labeled DNAJB1$^{154-341}$, Ydj1$^{111-351}$, and Ydj1$^{256-409}$ were grown at 37°C in M9 D$_2$O media supplemented with [$^2$H,$^{12}$C]-glucose and $^{15}$NH$_4$Cl as the sole source of carbon and nitrogen. In the case of DNAJB1, 2-ketobutyric acid-$^{13}$C$_4$,3,3-d$_2$ sodium salt (60 mg/L), 2-ketoisovaleric acid-$^{13}$C$_4$,d$_3$ sodium salt (80 mg/L), and $^{13}$C-L-methionine (100 mg/L) (Cambridge Isotope Laboratories) were added 1 hr prior to induction with 1 mM IPTG, following the procedure of *Tugarinov et al., 2006* to produce U-$^2$H, $^{15}$N, $^{13}$CH$_3$-ILVM labeled protein. Proteins were expressed at 25°C overnight.

### Purification of labeled and unlabeled proteins

Proteins were purified on a Ni-NTA HiTrap HP column (GE Life Sciences). The purification tag was cleaved by the appropriate protease (see Construct preparation), and the cleaved protein was further separated from the uncleaved protein, the tag, and the protease on a Ni-NTA HiTrap HP column. HSPB1, DNAJA2, and DNAJB1 chaperone variants were concentrated on an Amicon Ultra-15 10K molecular weight cutoff (MWCO) filter (Millipore) and further purified on a HiLoad 16/600 Superdex 200 pg gel filtration column (GE Healthcare), equilibrated with 25 mM HEPES pH 7.0, 150 mM KCl, and 2 mM DTT. Tau constructs were concentrated on an Amicon Ultra-15 3.5K MWCO filter (Millipore) and further purified on a HiLoad 16/600 Superdex 75 pg gel filtration column (GE Healthcare) equilibrated with 25 mM HEPES pH 7.0, 300 mM KCl, and 2 mM DTT. Purity of proteins was confirmed by SDS-PAGE.

### Aggregation-prevention assays

Aggregation kinetics were measured in Synergy H1 microplate reader (BioTek) in black, flat-bottom, 96-well plates (Nunc). Tau or tau$^{4R}$ variants (10 µM) were pre-incubated in the presence or absence of indicated chaperones for 10 min at 37°C. All proteins in the assay were buffer exchanged into the assay buffer (50 mM HEPES pH 7.4, 50 mM KCl, and 2 mM DTT). ThT (Sigma) at a final concentration

of 10 µM was added, and the aggregation was induced by the addition of 2.5 µM freshly prepared heparin salt solution (Sigma). Aggregation reactions were run at 37˚C with continuous shaking (567 rpm) and monitored by ThT fluorescence (excitation = 440 nm, emission = 485 nm, bandwidth), using an area scan mode with a $3 \times 3$ matrix for each well. Black, flat-bottom, 96-well plates (Nunc) sealed with optical adhesive film (Applied Biosystems) were used. For data processing, baseline curves at same conditions but without heparin were subtracted from the data. Samples were run in triplicate, and the experiments were repeated at least four times with similar results.

## Seeded tau aggregation reactions

Tau seeds were prepared from mature tau fibers generated under similar conditions to these in the aggregation-prevention assays, except that ThT was omitted. The fibers were then sonicated using a probe sonicator (Vibra-Cell, SONICS) with an amplitude of 40%, for 30 s on and 10 s off, for a total of 7 min. The sonicated fibers were immediately added to monomeric tau, ThT, and DTT in a 96-well plate in the ratios described above and ThT fluorescence was measured as a function of time.

## Dynamic light scattering (DLS)

The hydrodynamic radius of tau$^{4R}$ seeds was measured by DLS on a DynaPro DLS Plate Reader III (Wyatt Technology). Tau seeds (10 µM) were loaded on a 96-well black, clear-bottom plates (Nunc), and subjected to a 5 min 3000 $\times$ g centrifugation to remove air bubbles from the wells. Measurements were carried out 20 times per well before averaging, with 5 s acquisitions at 25˚C. Resulting autocorrelation functions were fitted with the equation

$$g_2(\tau) = 1 + \beta e^{-2Dq^2\tau} \tag{1}$$

where $\beta$ is the coherence factor, D is the translational diffusion coefficient, and q is the scattering wave vector given by

$$q = \frac{4\pi n}{\lambda_0} \sin\left(\frac{\theta}{2}\right) \tag{2}$$

where n is the solvent refractive index (n = 1.334 was used), $\lambda_0$ is the wavelength used by the instrument, and $\theta$ is the scattering angle.

The Stokes radius ($R_s$) was calculated from the translational diffusion coefficient, $D$, using the Stokes–Einstein equation

$$D = \frac{k_B T}{6\pi\eta R_s} \tag{3}$$

where $k_B$ is Boltzmann coefficient $\left(1.38 \cdot 10^{-23} \frac{kgm^2}{s^2 K}\right)$, T is the temperature (298 K), and $\eta$ is the dynamic viscosity of our buffer.

## Aggregation-prevention data fitting

All aggregation kinetics were fitted with a saturation-elongation-fragmentation model (*Meisl et al., 2016*) using a critical nucleus size of $n_c = 2$. The differential equation system for this model is

$$\dot{P}(t) = k_n m(t)^{n_c} + k_m M(t) \tag{4}$$

$$\dot{M}(t) = 2k_p \frac{K_E m(t)}{K_E + m(t)} P(t)$$

Here, $P(t)$ is the number concentration of fibrils, $M(t)$ is the mass concentration of a fibril, $m(t)$ is the monomer concentration, $k_n$ is the nucleation rate, $k_m$ is the fragmentation rate, $k_p$ is the elongation rate, and $K_E$ is the equilibrium constant for monomer addition to an existing fibril. The effect of heparin was not included explicitly in this model and is implicitly contained in the kinetic rates. For fitting, the ThT fluorescence signal $S_i(t)$ of the ith time trace was converted to the mass concentration of the fibrils $M_i(t)$ according to

$$M_i(t) = m_{max} \frac{S_i(t)}{S_{max}(\infty)} \tag{5}$$

Here, $m_{max}$ is the highest tau$^{4R}$ concentration used in the experiments and $S_{max}(\infty)$ is the ThT signal of the long-term plateau for the time trace with the highest tau$^{4R}$ concentration. The resulting mass concentration $M_i(t)$ was then fitted by numerically solving the differential equation system *Equation 4* for $M(t)$, using the initial conditions $M(0) = 0$, $P(0) = 0$, and $m(0) = m_i$, where $m_i$ is the initial concentration of tau$^{4R}$ monomers for the ith time trace. Prior to fitting, the time traces $M_i(t)$ were smoothed by binning data points to reduce noise and to speed up fitting. The bin size was 2–5 data points. Fitting was performed using the 'differential evolution' method (*Storn and Price, 1997*) in Mathematica 11.2 (Wolfram). Importantly, $k_p$ cannot be independently obtained from unseeded data. We therefore arbitrarily set $k_p = 1$ for the global fit of the unseeded data at all monomer concentrations, thus obtaining $k'_n = k_n k_p$ and $k'_m = k_m k_p$. In a second step, we determined $k_n$, $k_m$, and $k_p$ in a global fit of a data set including aggregation seeds using

$$\dot{P}(t) = \frac{k'_m}{k_p} M(t) \tag{6}$$

$$\dot{M}(t) = 2k_p \frac{K_E m(t)}{K_E + m(t)} P(t)$$

with the initial conditions $M(0) = M_0$, $P(0) = M_0/L$, and $m(0) = m_0 = 10\mu M$. Here, $M_0$ is the mass concentration of the seeds and $L$ is the length of the seeds. We determined $L$ by measuring the Stokes radius $R_s = 55nm$ of the seeds using DLS (see above). We then modeled the seeds as an ellipsoid with a long axis $a$ (seed length) and a short axis $b$ (fibril thickness). The friction coefficient for an ellipsoid is given by $\xi_e = 6\pi\eta a / \ln(2a/b)$, which must be identical to that of a sphere with the Stokes radius $R_s$ given by $\xi_s = 6\pi\eta R_s$. From the equality, the fibril length $a$ can be determined given that the fibril thickness $b$ is known. Based on existing cryo-EM structures of tau-fibrils (6QJH, 6QJM, 6QJP), we estimated $b \sim 10nm$, which results in $a \sim 200nm$. Given the spacing of tau monomers in a fibril of approximately 2 nm, we estimated a seed length of $L = 100$ tau monomers. Error estimates of the kinetic rates were obtained by fitting two independent data sets.

Fitting of the data in the presence of chaperones was performed for each kinetic trace individually by fixing the kinetic rates to those determined in the absence of chaperone and only allowing one rate to vary at a time. An exception was the data set for DNAJA2 in which we performed a simultaneous global fit of seeded and unseeded experimental traces (*Equations 4–6*) to identify the simultaneous effect of DNAJA2 on $k_n$, $k_p$, and $k_m$. For all fits in which only a single parameter was scanned, we used 'simulated annealing' to optimize the parameters. We would like to note that the amplitude of the aggregation kinetics was not a free-fitting parameter but was determined by the total concentration of tau$^{4R}$ monomers (10 μM) in the experiment. Hence, fitting of the aggregation kinetics in the presence of chaperones assumes that the presence of chaperone does not alter the ThT concentration accessible in solution to stain the fibrils.

## NMR spectroscopy

All NMR experiments were carried out at 25°C on 14.1 T (600 MHz), 18.8T (800 MHz), or 23.5 T (1000 MHz) Bruker spectrometers equipped with triple resonance single (z) or triple (x, y, z) gradient cryoprobes. The experiments were processed with NMRPipe (*Delaglio et al., 1995*) and analyzed with NMRFAM-SPARKY (*Goddard and Kneller, 2000*) and CCPN (*Vranken et al., 2005*).

## NMR assignment experiments

Assignments for tau$^{4R}$ were transferred from the BMRB (entry 19253) and corroborated by HNCACB, CBCA(CO)NH, HN(CA)CO, and HNCO experiments on a 4 mM sample of [U-$^{15}$N,$^{13}$C]-labeled tau$^{4R}$ in 50 mM HEPES pH 7.4, 50 mM KCl, 1 mM DTT, 0.03% NaN$_3$, and 10% D$_2$O. The assignment experiments were recorded on an 800 MHz magnet, resulting in the unambiguous assignment of 90% of non-proline residues.

Tau-heparin complex assignments were obtained by recording 3D HNCA, CBCA(CO)NH, and HN(CA)CO on a 2.5 mM [U-$^{13}$C,$^{15}$N]-labeled tau$^{4R}$ sample supplemented with 2.5 mM heparin. The experiments were recorded on an 800 MHz magnet and 88% of non-proline residues were assigned.

## Secondary structure propensities

Secondary structure propensities for tau$^{4R}$ and tau$^{4R}$-heparin complex were calculated from backbone C′, C$_\alpha$, and C$_\beta$, $^1$H, $^{15}$N chemical shifts following a procedure described in *Marsh et al., 2006*. Proline and cysteine residues were omitted from this calculation.

## Residual dipolar couplings (RDCs)

Backbone amide $^1$D$_{NH}$ RDCs were measured using a 300 µM sample of tau$^{4R}$, tau P301L, or tau P301S diluted in 50 mM HEPES pH 7.4 buffer with 100 mM KCl. The one-bond N-H RDCs were determined by using inphase-antiphase (IPAP)-HSQC experiments (*Ottiger et al., 1998*), and $^1$D$_{NH}$ values were calculated as the difference between splittings measured in the isotropic phase and in a sample in which tau$^{4R}$, tau$^{4R}$ P301S, or tau$^{4R}$ P301L had been aligned in 5 mg/mL Pf1 bacteriophage (Asla) or in 4.5% (vov/vol) C12E5/n-hexanol alignment medium (Sigma). The experiments were recorded on a 1000 MHz Bruker spectrometer and RDCs ranged from +20 to −10 Hz. The heparin-bound tau sample (1:1 tau:heparin molar ratio) was only aligned in 16 mg/mL bacteriophage pf1 and the measured RDCs ranged from +9 to −7 Hz.

## NMR tau-chaperone binding experiments

Tau interaction with chaperones was assayed for 200 µM samples of [U-$^{15}$N]-labeled tau, tau$^{4R}$, or tau$^{4R}$ P301S and P301L mutants. Tau variants were measured alone or upon addition of heparin (200 µM) and/or chaperones (100 or 400 µM; as indicated in spectrum) in 50 mM HEPES pH 7.0, 50 mM KCl, 1 mM DTT, 0.03% NaN$_3$, and 10% D$_2$O. $^1$H-$^{15}$N HSQC-TROSY spectra were acquired for each sample, and peak intensities were determined by quantifying peak volumes. Regions of tau$^{4R}$ with signal loss greater than one standard deviation from the average intensity ratio were determined to be the regions of binding.

High salt binding experiments were performed with samples of $^{15}$N-tau$^{4R}$ (200 µM) and 400 µM [U-$^1$H]-labeled chaperones in 50 mM HEPES pH 7.0, 300 mM KCl, 1 mM DTT, 0.03% NaN$_3$, and 10% D$_2$O. Binding was determined by calculating intensity ratios as described above.

The binding of DNAJB1 and DNAJA2 chaperones to tau$^{4R}$ was measured by acquiring $^1$H-$^{15}$N HSQC-TROSY spectra for 200 µM [U- $^2$H,$^{15}$N]-labeled DNAJB1$^{154-341}$ or Ydj1$^{111-351}$ alone or with twofold excess of deuterated tau$^{4R}$. The reactions were measured in 50 mM HEPES pH 7.0, 50 mM KCl, 1 mM DTT, 0.03% NaN$_3$, and 10% D$_2$O. Backbone DNAJB1$^{154-341}$ and Ydj1$^{111-351}$ assignments were available through the BMRB (entries 27998 and 28000, respectively).

The interaction of full-length DNAJB1 to tau$^{4R}$ was determined by acquiring $^1$H-$^{13}$C HMQC methyl-TROSY spectra (*Tugarinov et al., 2003*) for 100 µM [$^2$H, $^{13}$CH$_3$]-ILVM-labeled DNAJB1 alone or with 200 µM $^2$H-tau$^{4R}$ (or tau-heparin complex) in 50 mM HEPES pH 7.4, 100 mM KCl, 2 mM DTT and 0.03% NaN$_3$ in 100% D$_2$O. ILVM assignments for full-length DNAJB1 were taken from previous work in our lab (*Faust et al., 2020*). Binding regions were determined by intensity ratio as described above.

## NMR chemical shift perturbations

The interaction of tau$^{4R}$ with heparin was monitored by 2D $^1$H–$^{15}$N HSQC experiments. Heparin (40–400 µM) was titrated into 200 µM of $^{15}$N-labeled tau$^{4R}$ in 50 mM HEPES pH 7.0, 50 mM KCl, 1 mM DTT, 0.03% NaN$_3$, and 10% D$_2$O and chemical shifts were recorded.

CSPs were calculated from the relation

$$\Delta\delta = \sqrt{\Delta\delta_H 2} + \left(\frac{\Delta\delta_N}{5}\right)^2 \tag{7}$$

where $\Delta\delta_H$ is the amide proton chemical shift difference, and $\Delta\delta_N$ is the $^{15}$N backbone chemical shift difference. CSPs greater than one standard deviation from the mean were considered significant.

## Negative stain electron microscopy

Tau fibrils or tau fibrils-chaperone mixtures (10 µl) were deposited on glow-discharged carbon-coated copper EM grids (Electron Microscopy Sciences), washed with three consecutive drops of 1% w/v Uranyl-formate, and air-dried. Imaging was performed on an FEI T12 Spirit transmission electron microscope at 120 kV and a magnification of 9300–30,000 times, equipped with a Gatan OneView CMOS $4K \times 4K$ CCD camera.

## Chaperone-fibril co-sedimentation assay

Preformed tau$^{4R}$ fibers (10 µM) were incubated with HSPB1, DNAJB1, and DNAJA2 chaperones (10 µM) for 20 min at 37°C in 50 mM HEPES pH 7.4 and 50 mM KCl. Tau fibers were separated from the unbound chaperones by centrifugation at 16,900 g for 30 min. The pellets were washed, resuspended in 50 µL of buffer with 20% SDS, and sonicated for 10 min. Samples were incubated for 5 min at 95°C and run on a 4–20% gradient SDS-PAGE gel (GenScript).

## Fluorescence anisotropy measurements

Steady-state equilibrium binding of DNAJA2, DNAJB1 chaperones to preformed tau fibers was measured by fluorescence polarization using 100 nM of fluorescently tagged chaperones (DNAJB1 G194C-AF488 or DNAJA2-AF488).

Steady-state equilibrium binding of DNAJB1, DNAJA2, and HSPB1 to monomeric tau was measured by fluorescence anisotropy using 100 nM of fluorescently tagged tau (tau C291S, C322S, L243C-AF488). Samples were allowed to equilibrate for 10 min at 37°C, and measurements were performed on a Tecan SPARK 10M plate reader in black, flat-bottomed 384 square well plates. The excitation filter was centered on 485 nm with a bandwidth of 20 nm, and the emission filter was centered on 535 nm with a bandwidth of 25 nm. The gain and Z position were optimized from a well in the center of the binding curve, followed by calibration of the G factor. 60 flashes were performed per well.

Data were fit to a one-site binding model using OriginPro version 2018.

## Acknowledgements

RR is supported by the European Research Council starting grant (ERC-2018-STG 802001), Abisch Frenkel Foundation for the Promotion of Life Sciences, and a research grant from the Blythe Brenden-Mann New Scientist Fund. The electron microscopy studies were partially supported by the Irving and Cherna Moskowitz Center for Nano and BioNano Imaging (Weizmann Institute of Science). We thank Eran Ofek for fruitful discussions and advice.

## Additional information

### Competing interests

Rina Rosenzweig: Reviewing editor, *eLife*. The other authors declare that no competing interests exist.

### Funding

| Funder | Grant reference number | Author |
| --- | --- | --- |
| H2020 European Research Council | 802001 | Rina Rosenzweig |
| Abisch-Frenkel-Stiftung | | Rina Rosenzweig |
| Weizmann Institute of Science | | Sharon Grayer Wolf |

The funders had no role in study design, data collection and interpretation, or the decision to submit the work for publication.

## Author contributions
Rose Irwin, Conceptualization, Data curation, Formal analysis, Validation, Investigation, Visualization, Methodology, Writing - original draft, Writing - review and editing; Ofrah Faust, Conceptualization, Data curation, Formal analysis, Validation, Investigation, Methodology, Writing - original draft, Writing - review and editing; Ivana Petrovic, Conceptualization, Data curation, Formal analysis, Validation, Investigation, Visualization, Writing - original draft, Writing - review and editing; Sharon Grayer Wolf, Resources, Data curation, Formal analysis, Validation, Methodology, Writing - review and editing; Hagen Hofmann, Resources, Formal analysis, Supervision, Validation, Methodology, Writing - review and editing; Rina Rosenzweig, Conceptualization, Resources, Formal analysis, Supervision, Funding acquisition, Validation, Writing - original draft, Project administration, Writing - review and editing

## Author ORCIDs
Rose Irwin ⓘ https://orcid.org/0000-0002-1147-4944
Ofrah Faust ⓘ https://orcid.org/0000-0002-0105-5922
Ivana Petrovic ⓘ https://orcid.org/0000-0003-4461-7603
Sharon Grayer Wolf ⓘ http://orcid.org/0000-0002-5337-5063
Rina Rosenzweig ⓘ https://orcid.org/0000-0002-4019-5135

## Decision letter and Author response
Decision letter https://doi.org/10.7554/eLife.69601.sa1
Author response https://doi.org/10.7554/eLife.69601.sa2

## Additional files

### Supplementary files
• Transparent reporting form

### Data availability
All data generated or analyzed during this study are included in the manuscript and supporting files. Source data files are provided for Figures 1, 2, 3, and 4 including the full raw unedited gel in figure 2A.

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
