## [Decision Letter]

**Acceptance summary:**

This paper presents a series of elegant biophysical approaches to address the intriguing hypothesis that different molecular chaperones may recognize and bind distinct tau species, and thus may use different mechanisms to prevent tau aggregation. The findings advance our understanding of how chaperones can counteract the deleterious effect of tau amyloidogenesis providing important insights into tau aggregation suppressors that will inform future cell based experiments.

**Decision letter after peer review:**

Thank you for submitting your article "Hsp40s play complementary roles in the prevention of tau amyloid formation" for consideration by *eLife*. Your article has been reviewed by 3 peer reviewers, and the evaluation has been overseen by a Reviewing Editor and Volker Dötsch as the Senior Editor. The following individual involved in review of your submission has agreed to reveal their identity: Charalampos Babis Kalodimos (Reviewer #1).

Essential revisions:

1) It is proposed that the differential effects resemble either exposure or reduced accessibility of the PHF6 regions. How can DNAJA2 induce NMR spectral changes to the PHF6 region in the non-aggregation-prone tau form, where these regions are proposed to be inaccessible? Does this reflect binding to monomers?

2) Figure 3 and model fitting: Are the various fittings statistically significant i.e. in Figure 3C an apparent better fitting is found then considering effects on kn and kp, – could this simply reflect allowing an additional parameter/degree of freedom? Moreover, Figure 3-Figure supp 1 is not clear. What is HMWH? Was heparin added? In what concentration? How is that considered when modeling the influence of seeded aggregation? Not knowing how aggregation was induced makes it hard to judge the procedure. It is also not clear what data are plotted in panel A. Is it from panel B or C? The positive curvature of the scaling exponent seems to rely on a single point (the last one). The error bar on this point is compatible with a linear behavior. In addition, I don't see a clear support of the following statement "the scaling exponent was determined to be -0.34 {plus minus} 0.04, which is consistent with a dominant primary nucleation pathway and a contribution stemming from the presence of fibril fragmentation". It should be described in more details, in particular because it is the base to interpret the fitting of chaperon-inhibited aggregation.

3) Figure 4: How are the differential effects of heparin and P301S tau species seen upon binding of DNAJB1 and HSPB1 rationalized? The authors suggest that both conditions expose PHF6 motifs, but why are then different interactions observed in the presence of heparin or the P301S mutation? Can the authors provide more information about the structural features of the conformational ensembles in either case? For Figure 4, supplement 1: the authors should show the direct secondary chemical shift differences rather than interpolated SSPs. In any case, there are some differences seen compared to Tau alone and in the presence of heparin, also in the PHF6 regions, that the authors may comment on.

4) The authors have mainly used NMR line-broadening to report on molecular interactions, which can report on the interfaces proposed but could also involve aggregation of the proteins involved. Can the authors rule out such affects? Also, NMR intensity ratios I/I_0 should indicate error bars, at least for some representative examples.

5) The authors have used monomeric variants of Hsp40. What are the thoughts on the role of dimeric interactions mediated by the two Hsp40s? How does this affect the proposed interactions and model.

6) It is difficult to reconcile the results from Figure 3 and from Figure 5 for HSPB1. If HSPB1 inhibits aggregation mostly by slowing elongation, how come the fiber length is unaffected? If it was acting on nucleation, I would expect fewer fibrils of the same length, but if it is acting on elongation, I would expect large numbers of fibrils with smaller length. The explanation given in the results is not convincing to me: "HSPB1 thus appears to only affect the number of fibrils, not the fibril length itself, indicating that its interaction with tau monomers merely slows the fibril elongation rates, but does not prevent their incorporation into the amyloid fibers".

7) It seems to be small differences in the interaction regions in tau4RD and 2N4R (Figure 1 supp). From the assignment shown, the region seems to extend up to 318 for 2N4R. This difference would be worth describing and maybe discussing. In the method, it is not mentioned how much assignment was achieved for 2N4R. If this number is low, does it change the confidence in the interaction region of 2N4R?

8) It is very surprising that addition of the DNAJB1 or DNAJA2 to mature fibrils change so drastically the morphology, seemingly from curly to straight morphology. Could the authors comment on that?

9) It is mentioned that 88% of non proline residues were assigned. Yet in figure 1D, all residues seem to have an intensity value. It would be more precise not to show an intensity reduction for the residues that could not be assigned.

---

## [Author Response]

Essential revisions:1) It is proposed that the differential effects resemble either exposure or reduced accessibility of the PHF6 regions. How can DNAJA2 induce NMR spectral changes to the PHF6 region in the non-aggregation-prone tau form, where these regions are proposed to be inaccessible? Does this reflect binding to monomers?

We thank the reviewers for bringing up this important point. Recent work (by Mirbaha et al., doi: 10.7554/eLife.36584, and Chen at al., doi: 10.1038/s41467-019-10355-1) has shown that inert tau monomers can adopt transient β-hairpin structures that partially shield the aggregation-prone PHF6 motifs. Indeed, our RDC measurements confirm the presence of such a compaction, with large RDCs measured in the PHF6 repeats region (Figure 4 —figure supplement 3). However, even in this more compacted tau state, MD simulation and CS-Rosetta structures of this tau species (Mirbaha et al., doi: 10.7554/eLife.36584) show that the PHF6 hydrophobic side chains are still accessible and available to the chaperones. Thus, we hypothesize that HspB1 chaperone and DNAJA2 (via its CTDI domain), which are known to interact with the hydrophobic side chains in peptides, can bind to this monomeric tau state.

Interestingly, we find that the CTDII sites of both DNAJB1 and DNAJA2 chaperones do not recognize this compact monomeric tau state, but can rather bind with high affinity to the extended tau conformation. While further studies (outside the scope of this manuscript) will be required to fully understand the structural preferences in the CTDII domains of DNAJA2 and DNAJB1, there are several possible options – (1) the binding to CTDII domains involves backbone residues that only become exposed in the expanded tau conformation, or (2) the PHF6 domains need to interact simultaneously with the two CTDII domains in the DNAJA2 and DNAJB1 dimers, which can only take place in the expanded PHF6 conformation, or rather (3) additional residues required for the binding only become accessible in the aggregation-prone expanded tau conformation.

2) Figure 3 and model fitting: Are the various fittings statistically significant i.e. in Figure 3C an apparent better fitting is found then considering effects on kn and kp, – could this simply reflect allowing an additional parameter/degree of freedom?

This is indeed a very important point and we thank the reviewers for bringing it up. To validate that the DNAJA2 chaperone indeed affects several rates with higher accuracy, we have recorded an additional set of kinetic experiments measuring the aggregation prevention by DNAJA2 chaperones on seeded tau aggregation. The idea behind these seeded experiments is that under seeding conditions, the primary nucleation events are negligible, allowing us to obtain a better precision in determining the effect of the chaperone on other rates such as elongation. We therefore globally fit this substantially increased data set of seeded and unseeded experiments in the presence of DNAJA2 chaperone with three degrees of freedom (nucleation rates, elongation rates, and fragmentation rates).

The results from the new experiments (shown in revised manuscript figure 3 —figure supplement 2) show with statistical significance that DNAJA2 chaperone indeed reduces the rates of tau elongation and nucleation.

Moreover, Figure 3-Figure supp 1 is not clear. What is HMWH? Was heparin added? In what concentration? How is that considered when modeling the influence of seeded aggregation? Not knowing how aggregation was induced makes it hard to judge the procedure. It is also not clear what data are plotted in panel A. Is it from panel B or C?

Following the reviewers’ comments regarding the legend of Figure 3-Figure supp 1, the term HMWH, which referred to high molecular weight heparin, has been corrected to heparin. The legend now also includes the concentrations of heparin used (0.5:1 molar ratio heparin to monomeric tau in all the unseeded reactions).

We have likewise added a detailed description to the methods section of the procedure used for fitting the aggregation kinetics, both for the seeded and unseeded tau.

The positive curvature of the scaling exponent seems to rely on a single point (the last one). The error bar on this point is compatible with a linear behavior. In addition, I don't see a clear support of the following statement "the scaling exponent was determined to be -0.34 {plus minus} 0.04, which is consistent with a dominant primary nucleation pathway and a contribution stemming from the presence of fibril fragmentation". It should be described in more details, in particular because it is the base to interpret the fitting of chaperon-inhibited aggregation.

We agree with the reviewer on this and have therefore added an additional measurement point of 40 μm monomeric tau to the plot, which better shows the appearance of the positive curvature. It is, however, important to note that the mechanism of tau aggregation has been previously reported to consist of the saturating elongation step (which is indicated by the positive curvature in the double logarithmic plots), and our current results only serve to confirm that this is also the case under our experimental conditions.

As for the scaling exponent, this was calculated from the half-times of tau aggregation at various concentrations, and the tau aggregation mechanism was determined based on the procedure published by Tuomas Knowles’ group in Nature protocols (https://doi.org/10.1038/nprot.2016.010). The reported scaling factor corresponds to the initial, linear part of the curve, excluding the points showing the positive curvature. We have added a detailed explanation of the procedure both in the main text and in the figure legend of Figure 3 – figure 1 supplement 1 of the revised manuscript:

“To extract the scaling factor (γ), the half-times (t1/2) of the aggregation reaction were plotted versus the tau monomer concentration on a double logarithmic plot with the slope of this plot providing the scaling exponent (log(t1/2) = γlog(m0) + constant). By using the rate laws for the time evolution of aggregate mass, γ can be related to the reaction orders for each of the models. The scaling factor for tau aggregation was calculated to be -0.34±0.04, with -0.5<γ<0 indicating primary nucleation and fragmentation as the correct model. Furthermore, the deviation of the points from a straight line shows that γ is dependent on the monomer concentration, with positive curvature indicating the presence of saturating elongation (45).”

3) Figure 4: How are the differential effects of heparin and P301S tau species seen upon binding of DNAJB1 and HSPB1 rationalized? The authors suggest that both conditions expose PHF6 motifs, but why are then different interactions observed in the presence of heparin or the P301S mutation? Can the authors provide more information about the structural features of the conformational ensembles in either case?

While both heparin binding and the P301S mutation result in the increase of the “aggregation-prone” conformation of tau, with the more expanded PHF6 motifs, the two species differ dramatically in their populations of this conformation. In the case of heparin-bound tau, the aggregation-prone, expanded species is the major, highly-populated state of the protein, while the P301S/L mutations merely cause an increase in the population of the expanded conformers within the structural ensemble (as reported by Chen at al., doi: 10.1038/s41467-019-10355-1 and Kawasaki and Tale doi.org/10.3390/ijms21113920). Based on previous publications and our RDC measurements, in the mutants this expanded species represents just a small percentage (~15%) of the overall ensemble, and therefore only interacts with DNAJB1 weakly (although significantly better than the WT tau). The remaining ~90%, however (the inert tau population), can still readily bind to HSPB1.

For Figure 4, supplement 1: the authors should show the direct secondary chemical shift differences rather than interpolated SSPs. In any case, there are some differences seen compared to Tau alone and in the presence of heparin, also in the PHF6 regions, that the authors may comment on.

We have added a secondary chemical shifts plot to Figure 4 —figure supplement 1 (panel D), which does not show any significant changes in the secondary structure propensities between tau and heparin-bound tau species.

4) The authors have mainly used NMR line-broadening to report on molecular interactions, which can report on the interfaces proposed but could also involve aggregation of the proteins involved. Can the authors rule out such affects?

In order to rule out the presence of aggregation, we ran a set of DLS experiments conducted at identical protein concentrations and conditions as our NMR binding experiments. We did not observe any changes in particle size for tau-chaperone complexes over the course of 4 hours, indicating that these do not aggregate in the course of our NMR measurements. Furthermore, DLS measurements of tau and tau-heparin samples also showed that these proteins remain stable during our measurements and do not aggregate.

Also, NMR intensity ratios I/I_0 should indicate error bars, at least for some representative examples.

We have calculated the error bars for all tau-chaperone NMR binding experiments. These are added as source data for figures 1 and 4 in the revised manuscript.

5) The authors have used monomeric variants of Hsp40. What are the thoughts on the role of dimeric interactions mediated by the two Hsp40s? How does this affect the proposed interactions and model.

We thank the reviewers for bringing up this interesting point. We now make it more clear in the manuscript that in all biochemical experiments and NMR experiments with 15N-labeled tau we used the full length versions of the chaperones. Also all the affinities reported in the paper were likewise acquired with these full length, dimeric versions of the proteins. The truncated variant was used only when characterizing the interaction of tau with labeled DNAJA2 chaperone (i.e., from the chaperone side), where a monomeric version of DNAJA2 yeast homologue Ydj1 was used.

We have added an explicit explanation to the text of the revised manuscript regarding the truncated construct used.

Furthermore, in order to ensure that the interaction of the aggregation-prone tau, which we mapped to the CTDII region of Ydj1, was not affected by the lack of the dimerization domain and C-terminal tail of the chaperone, we have repeated the binding experiments using the dimeric form of the DNAJA2 CTDII domains (CTDII+DD construct – residues 256-409). The construct showed significant chemical shift perturbation upon binding to tau-heparin, but not to tau alone, confirming that the aggregation-prone species indeed interacts with DNAJA2 CTDII.

6) It is difficult to reconcile the results from Figure 3 and from Figure 5 for HSPB1. If HSPB1 inhibits aggregation mostly by slowing elongation, how come the fiber length is unaffected? If it was acting on nucleation, I would expect fewer fibrils of the same length, but if it is acting on elongation, I would expect large numbers of fibrils with smaller length. The explanation given in the results is not convincing to me: "HSPB1 thus appears to only affect the number of fibrils, not the fibril length itself, indicating that its interaction with tau monomers merely slows the fibril elongation rates, but does not prevent their incorporation into the amyloid fibers".

We thank the reviewers for this comment, which brought to our attention the inadequate explanation of the results observed in figure 5 in our original manuscript.

Reanalyzing of EM images showed that the addition of HSPB1 chaperone to the aggregation reaction does result in a minor decrease of fibril length, however one that is significantly smaller to what was observed in the presence of the J-domain proteins. This small reduction is also consistent with our aggregation kinetics measurements, which showed only a small (16%) reduction of fibril mass upon addition of HSPB1 chaperone at 0.25:1 HSPB1:tau ratio. However, since fibril mass decrease can result from either reduction in fibril length or from an overall decrease of the number of fibrils, this can only serve as an upper bound for the reduction in fibril length. Regardless, such small changes in length are difficult to accurately quantify from the EM images, leading to our assumption that HSPB1 does not result in changes to fibril length. Even the addition of 0.75:1 ratio of HSPB1:tau only decreased fibril mass by 37%, which is significantly smaller effect than that of the DNAJA2 and DNAJB1 chaperones, which reduced fibril mass by 91% and 87%, respectively. Due to these diminished reductions, identifying the specific effects of HSPB1 on fibril length would require analysis outside the scope of this manuscript.

Overall, HSPB1 chaperone only has a minor effect on slowing down fiber formation, which is a result of both the weak interaction of the chaperone solely with the monomeric tau species, and its limited ability to reduce the rate of tau elongation into the growing fibers.

New plots, showing the decrease in fibril mass as a function of chaperone concentration are now included as Figure 5 —figure supplement 1 in the revised manuscript. Furthermore, the section describing the “Changes to tau fibers caused by the chaperones” has been rewritten to incorporate the points discussed above:

“Since ThT fluorescence only reports on total fibril mass, with no differentiation to fibril length or number, we turned to EM in order to image the fibers. […] Thus, the weak interaction of HSPB1 with tau monomers along with its limited ability to only slow down the fibril elongation rates, is not sufficient to efficiently prevent tau incorporation into the amyloid fibers.”

7) It seems to be small differences in the interaction regions in tau4RD and 2N4R (Figure 1 supp). From the assignment shown, the region seems to extend up to 318 for 2N4R. This difference would be worth describing and maybe discussing. In the method, it is not mentioned how much assignment was achieved for 2N4R. If this number is low, does it change the confidence in the interaction region of 2N4R?

Indeed, when adding HSPB1 to tau 2N4R, the peak broadening extends also to residues 314-318. This broadening of the extra 4 residues could possibly be attributed to the larger size, and therefore slower tumbling times, of the full length tau compared to tau 4RD.

It is important to note, however, that all the residues in tau 2N4R that show peak broadening upon binding to HSPB1 and DNAJA2 have been assigned and were found to be located in the PHF6 regions.

8) It is very surprising that addition of the DNAJB1 or DNAJA2 to mature fibrils change so drastically the morphology, seemingly from curly to straight morphology. Could the authors comment on that?

We were also very intrigued by this drastic change in fiber appearance. After re-reading the manuscript, though, we think that the term “morphology” may be a bit misleading. We cannot determine whether the addition of the chaperones indeed modifies the fiber morphology, or whether this is just due to the fiber mobility being restricted as a result of the chaperones binding densely on the fiber, thereby changing their appearance to less twisted, straighter fibers. We have therefore changed the term “morphology” to “appearance” in the revised version of the manuscript

9) It is mentioned that 88% of non proline residues were assigned. Yet in figure 1D, all residues seem to have an intensity value. It would be more precise not to show an intensity reduction for the residues that could not be assigned.

The unassigned residues were never included in our intensity plots, but rather the numbering was non-continuous in our figures. We have now included plots, along with a full table of intensities, that specifically indicate the unassigned residues for all tau-chaperone NMR binding experiments.

These have been added as source data for figures 1 and 4 in the revised manuscript.